# Reparametrization mode Ward Identities and chaos in higher-pt. correlators in CFT$_2$

**Arnab Kundu**$^{a,1}$, **Ayan K. Patra**$^{a,2}$, **Rohan R. Poojary**$^{b,3}$

$^a$*Theory Division, Saha Institute of Nuclear Physics,*
*Homi Bhaba National Institute (HBNI),*
*1/AF, Bidhannagar, Kolkata 700064, India.*

$^b$*Institute for Theoretical Physics, TU Wien,*
*Wiedner Hauptstrasse 8-10, 1040 Vienna, Austria.*

*E-mail:* $^1$ arnab.kundu@saha.ac.in, $^2$ ayan.patra@saha.ac.in, $^3$ rpglaznos@gmail.com

ABSTRACT: Recently introduced reparametrization mode operators in CFTs have been shown to govern stress tensor interactions *via* the shadow operator formalism and seem to govern the effective dynamics of chaotic systems. We initiate a study of Ward identities of reparametrization mode operators *i.e.* how two dimensional CFT Ward identities govern the behaviour of insertions of reparametrization modes $\epsilon$ in correlation functions: $\langle \epsilon\epsilon\phi\phi \rangle$. We find that in the semi-classical limit of large $c$ they dictate the leading $\mathcal{O}(c^{-1})$ behaviour. While for the 4pt function this reproduces the same computation as done by Heahl, Reeves & Rozali in [1], in the case of 6pt function of pair-wise equal operators this provides an alternative way of computing the Virasoro block in stress-tensor comb channel. We compute a maximally out of time ordered correlation function in a thermal background and find the expected behaviour of an exponential growth governed by Lyapunov index $\lambda_L = 2\pi/\beta$ lasting for twice the scrambling time of the system $t^* = \frac{\beta}{2\pi} \log c$ for the maximally braided type of *out-of-time-ordering*. However when only the internal operators of the comb channel are *out-of-time-ordered*, the correlator sees no exponential behaviour despite the inclusion of the Virasoro contribution. From a bulk perspective for the *out-of-time-ordered* 4pt function we find that the Casimir equation for the stress tensor block reproduces the linearised back reaction in the bulk.

## 1  Introduction

Thermodynamics provides us with a universal infra-red (IR) description of a remarkably wide range of systems: across the scale of elementary particles to the large-scale structure of the observed Universe. Chaotic dynamics underlies the dynamical process towards thermalization in generic systems with a large number of degrees of freedom. Therefore, understanding better the role of chaos and thermalization in dynamical systems, in general, is an aspect of ubiquitous importance across various disciplines of physics.

Such questions have mostly been addressed conventionally in the framework of various spin-systems, and the likes. In recent years, however, in resonance with other advances in the framework of quantum field

theory and quantum gravity, such questions have found an important place in understanding salient features of quantum nature of gravity. Conformal Field Theories (CFTs), particularly in two-dimensions, provide us with a remarkable control with which such dynamical aspects can be probed and explored in a precise sense. The CFT-intuition becomes a cornerstone of understanding these aspects in both the framework of QFT (e.g., in a perturbative deformation of a CFT), as well as in quantum gravity (primarily via Holography).

Motivated with this broad perspective, we explore further the chaotic dynamics in a two-dimensional CFT, with a large central charge and a sparse spectrum. Usually, it is assumed[1] that such CFTs have a Holographic dual and are therefore related to the quantum properties of a black hole in an anti-de Sitter (AdS) space-time. The gravity answer is reproduced assuming the dominance of the identity block over blocks of other operators in the theory and maximising over all possible exchanges [3–5]. In a unitary CFT, any primary field fuses with itself via the identity operator and therefore the corresponding identity block contributes universally to any correlator where such operator fusion occurs. Particularly, the stress-tensor, which is a descendant of the identity in 2d plays an important role in the identity block. In a holographic setting these correspond to graviton exchanges in the bulk and $AdS_3$ physics can be gleaned out from studying different kinematical regimes [6–11]. These therefore exhibit the typical expected behaviour of out of time ordered correlation functions in a thermal state of an exponential growth[12–17], which in the bulk corresponds to scrambling caused by exchange of gravitons close to the horizon of the black hole [18].

In the context of $AdS_3$/$CFT_2$ boundary gravitons are the only degrees freedom and their effective theory when coupled with other fields can stand for a generating function of stress tensor interactions in the CFT. In this context the theory of "reparametrization modes" was expanded upon to study the contributions of identity blocks in $CFT_2$ [19–21]. This was explored in detail by Cotler and Jensen[22] by carefully considering phase space quantization [23, 24] of boundary gravitons in $AdS_3$ with relevant boundary conditions and was found to be governed by 2 copies of the Alekseev-Shatashvili action- found by path integral quantization of the co-adjoint orbit of $\text{Diff}(\mathbb{S}^1)/\text{SL}(2,\mathbb{R})$.[2] This is a theory of reparametrizations sourced by the boundary gravitons. This theory also consisted of bi-local operators the 2pt functions of which encoded contributions of the Virasoro blocks in a perturbative expansion in $1/c$ about large a central charge $c$. Consequently the vacuum block contribution to 4pt operators was obtained in the LLLL & HHLL limit[3] consistent with previously known results [9, 26–28].

Global blocks corresponding to any operator exchange have been understood in terms of shadow operator formalism. It was found in [1] that the shadow operator formalism [29, 30] can be used to recast this theory at the linear level in terms of "reparametrization modes"- $\epsilon^\mu(x)$; descendant of which is the shadow of the stress-tensor. This was then used to obtain a succinct expression for the 4pt vacuum block in even dimensions; while in $d = 2$ it reproduced the single graviton exchange answer in the limit of light operator dimensions. The leading correction in $1/c$ to the 4pt functions of light operators is basically obtained from the 2pt functions of the bi-locals which are linear in $\epsilon^\mu(x)$, without having the need to perform any conformal integrals. These $\epsilon^\mu(x)$ operators therefore seem to capture the universal physics of chaotic behaviour as it is the 2pt functions of their bi-locals which grows exponentially when their temporal positions are out of time ordered. Such operators as $\epsilon^\mu$ have also made an appearance in describing the effective action of SYK like models close to criticality [31] and nearly $AdS_2$ geometries in JT theories describing near horizon dynamics of near extremal black holes [32, 33]. It is the Schwarzian action cast in terms of these $\epsilon^\mu$ operator that captures the interaction of bulk scalars with the near horizon $AdS_2$ thus exhibiting the chaotic behaviour associated with black holes. Unlike black holes in $AdS_{d>3}$ the analysis in $AdS_3$ around a BTZ geometry of such operators- although from a bulk perspective, allows one to explore dynamics of chaos far from extremality [34, 35].

In [1, 36], the role of stress-tensor exchanges and correspondingly the importance of the reparametrization modes were highlighted and the computational simplifications they render where made use of in determining the Identity block contributions to the four-point and six-point "star channel" functions in $CFT_2$, respectively.

---

[1]See *e.g.* [2].

[2]See also [25] for a previous work using geometric quantization.

[3]Where a pair of operators have dimensions that scale as $c$ while the others are small compared to $c$.

As usual, these computations are carried out in the Euclidean framework and then analytically continued to obtain the corresponding Lorentzian correlators. Of particular focus is the maximally braided out-of-time-order correlator (OTOC) that exhibits an exponentially growing mode in time-scales larger than the dissipation-scale [19].

As already mentioned, in [36], the reparametrization mode formalism was heavily used to unpack the physics mentioned above. Furthermore, additional assumptions were made in [36], about the structure of the six-point block, in order to explicitly compute the six-point function. Here the reparametrization mode formalism in $CFT_2$ was used in conjunction with a proposal for the non-linear version of the corresponding block in the "star" channel. The star channel is a natural generalization of the identity block considered in the 4pt case as all internal lines are those of the stress-tensor [37–39]. This channel can be contrasted with the well studied "comb" channel where the internal operators are scalar primaries[40–48]. To obtain the full contribution from the Virasoro blocks in this channel a non-linear realization of the bi-locals of $\epsilon^\mu$ had to be used, motivated from the works of Cotler and Jensen [22] and [1, 21]. This non-linear proposal is essential in capturing the non-linear interactions in the dual gravitational description, that is ultimately responsible for the behaviour of the corresponding correlator. It is plausible that the reparametrization mode formalism, fused with standard symmetry constraints involving the stress-tensor e.g. Ward identities, provides us with a powerful control on generic n-point correlator and its chaotic behaviour.

In this article, we initiate a study of how 2d Ward identities- of the form of $\langle T\phi\phi \rangle$, $\langle TT\phi\phi \rangle$ can be used in conjunction with the reparametrization modes and how it provides a powerful method of computing the higher point correlators. The main benefit of this approach is that the Ward identities themselves incorporate the non-linear interactions, allowing us to compute Virasoro contribution to the 6pt comb channel consisting of 3 pairs of pairwise identical operators.

From the bulk holographic perspective, it is interesting to understand the implications of these Ward identities on the reparametrization modes. We leave such Holographic aspects for future, however, we point out that the conformal Casimir equation, satisfied by the stress-tensor conformal block, reproduces the (source-less) linearized Einstein equation from a three-dimensional bulk perspective.

This paper is sectioned as follows: In section-2 we briefly review the shadow operator formalism. In section-3 we compute the $\langle \epsilon\epsilon\phi\phi \rangle$ correlator from the stress tensor Ward identity for $\langle TT\phi\phi \rangle$. We then argue how this correlator allows us to compute the Virasoro contribution to the 6pt stress-tensor comb channel of pair-wise equal operators with conformal dimensions $h \sim \mathcal{O}(c^0)$. The global answer for which is generally known for arbitrary comb channel with arbitrary operators [49]. We are also able to compute similar Virasoro contributions to simpler 8pt. and 9pt. generalizations of the 6pt. stress tensor comb channel algebraically. Here the 8 & 9 pt. functions are obtained by considering identical triplets instead of identical pairs of operators in the 6pt. function. In section-4 we study the behaviour of the Virasoro 6pt stress-tensor comb channel correlator for various *out-of-time-orderings* and find unlike the vacuum block of the 6pt star channel the Virasoro 6pt stress-tensor comb channel behaves similar to its global analogue.1 In section-5 we make some observations which yield some insight into the bulk perspective for 4pt OTOC.

The former version of this paper assumed that the reparametrization Ward identity for $\langle \epsilon\epsilon\phi\phi \rangle$ computed the leading order answer in $1/c$ to the star-channel vacuum block first computed in [36]. However detailed comparison with the result obtained in [36] showed this not to be the case and we expressly thank the referees of Sci-Post to have insisted on this check. Related to the above mistaken assumption the former version also consisted of attempts of writing the projector onto the vacuum blocks to arbitrary order in $1/c$ in terms of the conformal integrals of the reparametrization modes; which although an interesting avenue for future investigations have also been removed from this version. We also thank the referees for making important suggestions towards improving the rigour and presentation of the paper.

**Notation:** We make use of (lower-case)Greek alphabets to indicate $d$-dimensional vector indices. Lower-case barred and unbarred Roman alphabets indicate $d+2$-dimensional embedding space vector indices while $X, Y, Z$ indicate embedding space vectors. 2-dimensional coordinates are the standard (anti)holomorphic indicated by $(\bar{z})z$, $(\bar{x})x$, *etc.*. Shadow operators are indicated by an overhead $\sim$. We also use the short hand of the form $\phi(z_1) \equiv \phi_1$ or $\phi(x_1^\mu) \equiv \phi_1$, $T^{\mu\nu}(x_0) \equiv T_0^{\mu\nu}$ or $T_{\mu\nu}(x_0) \equiv T_{\mu\nu}^0$, $I^{\mu\nu}(x_1 - x_2) = I^{\mu\nu}(x_{12}) \equiv I_{12}^{\mu\nu}$ or $I_{\mu\nu}(x_{12}) \equiv I_{\mu\nu}^{12}$, bi-local dependence as $\mathcal{B}(x_1, x_2) = \mathcal{B}_{12}$ etc. for simplicity of notation along with similar use of negative numbers *i.e.* $\phi_{-1} = \phi(z_{-1})$ or $\phi_{-1} = \phi(x_{-1}^\mu)$. We further make use of the short hand $\langle \phi_1 \phi_2 \rangle \equiv \langle \phi_{1,2} \rangle$, $\langle \phi_1 \phi_2 \phi_3 X_3 X_4 X_5 \rangle \equiv \langle \phi_{1,2,3} X_{3,4,5} \rangle$. We also use $\int d^d x_0 = \int_0$ to indicate conformally invariant integrals[4] to save space.

## 2 Review of shadow operators and reparametrization modes.

In this section we review the shadow operator formalism and the use of reparametrization modes as described in [21]. We give a brief review of this formalism in $d$-dimensions as some aspects of it may be useful in solving for the conformal integrals. We also review this formalism in 2d as the analysis in this paper would be concerning CFT$_2$.

### 2.1 Reparametrization modes in CFT$_d$

The shadow of an operator of dimension $\Delta$ and spin $l$ is defined using the embedding space conformal integral as follows

$$\widetilde{\mathcal{O}}(y)^{\mu_1 \cdots \mu_l} = \frac{k_{\Delta, l}}{\pi^{d/2}} \int D^d x \, \frac{I(x-y)^{\mu_1 \nu_1} \cdots I(x-y)^{\mu_l \nu_l}}{((x-y)^2)^{d-\Delta}} \mathcal{O}_{\nu_1 \cdots \nu_l}(x)$$

$$\text{with,} \quad I_{\mu\nu}(x) = \eta_{\mu\nu} - 2\frac{x_\mu x_\nu}{x^2}$$

$$\text{where} \quad k_{\Delta, l} = \frac{\Gamma(\Delta - 1)\,\Gamma(d - \Delta + l)}{\Gamma(\Delta + l - 1)\,\Gamma(\Delta - \frac{d}{2})}. \tag{2.1}$$

For a primary scalar operator $\mathcal{O}$ with dim $\Delta$ it's shadow is therefore

$$\widetilde{\mathcal{O}}(X) = \frac{k_{\Delta, 0}}{\pi^{d/2}} \int D^d Y \, \frac{\mathcal{O}(Y)}{(-2X.Y)^{d-\Delta}}, \tag{2.2}$$

with $k(\Delta, l)$ as in (2.1) with $l = 0$. Here the embedding space coordinates $X^a = \{X^+, X^-, X^\mu\}$ on $\mathbb{R}^{d+1,1}$ wherein the CFT$_d$ is defined on an $\mathbb{R}^d$ with coordinates $x^\mu$. We refer to the work by Simmons-Duffins [29] for defining these conformal integrals. We note certain useful results in Appendix A.

Here we have abused the notation and not used embedding space coordinates in defining the integrals[5]. The shadow operators; though fictitious are useful in construction of the projectors which project onto the conformal block of the operators. Projectors from shadow operators are obtained by constructing

$$|\mathcal{O}| = k'(d, \Delta, l) \int d^d x \, \widetilde{\mathcal{O}}(x)|0\rangle\langle 0|\mathcal{O}(x). \tag{2.3}$$

The use of these projectors in 4pt functions yields not only the block corresponding to the operator $\mathcal{O}$ but also its shadow $\widetilde{\mathcal{O}}$

$$\frac{\langle \phi_1 \phi_2 | \mathcal{O} | \phi_3 \phi_4 \rangle}{\langle \phi_1 \phi_2 \rangle \langle \phi_3 \phi_4 \rangle} = C_{\phi\phi\mathcal{O}}^2 (G_\Delta^{(l)}(u, v) + G_{(d-\Delta)}^{(l)}(u, v))$$

---

[4]These are the only form of integrals we would encounter unless mentioned otherwise.
[5]We denote by $\int d^2 x$ as the 2dim conformally invariant integral for the rest of the paper.

$$\text{as } u \to 0, v \to 1 \quad G_\Delta^{(l)}(u,v) \sim u^{\frac{\Delta-l}{2}}(1-v)^l, \qquad G_{d-\Delta}^{(l)}(u,v) \sim u^{\frac{d-\Delta+l}{2}}(1-v)^l \tag{2.4}$$

where $u = \frac{x_{12}^2 x_{34}^2}{x_{13}^2 x_{24}^2}$, $v = \frac{x_{14}^2 x_{23}^2}{x_{13}^2 x_{24}^2}$ are the invariant cross ratios. Here $G_\Delta^{(l)}$ is the conformal block of the operator $\mathcal{O}$ and $G^{(l)d-\Delta}$ of its shadow $\widetilde{\mathcal{O}}$. In order to isolate the conformal block contribution of $\mathcal{O}$ we need to subtract out the the contribution coming from the shadow block by discerning its behaviour as $u \to 0$. For example, as was seen in [1], in the 2d case this is simply achieved by imposing holomorphic factorization in $\{z, \bar{z}\}$. The reparemetrization modes introduced in [1] are defined using the shadow $\widetilde{T}$ of the stress-tensor

$$\widetilde{T}_{\mu\nu}(x) = \frac{2C_T \pi^{d/2}}{k_{0,2}} \mathbb{P}_{\mu\nu}^{\alpha\beta} \, \partial_\alpha \epsilon_\beta, \qquad \mathbb{P}_{\mu\nu}^{\alpha\beta} = \tfrac{1}{2}\left(\delta_\mu^\alpha \delta_\nu^\beta + \delta_\nu^\alpha \delta_\mu^\beta\right) - \tfrac{1}{d}\eta^{\alpha\beta}\eta_{\mu\nu} \tag{2.5}$$

where $\mathbb{P}_{\mu\nu}^{\alpha\beta}$ is a projector onto traceless and symmetric part and $C_T = c/2$ is the coefficient of the stress-tensor 2pt function with $c$ the central charge . It is worth noting that as the shadow $\widetilde{T}$ has conformal weight $\Delta = 0$ the reparametrization mode operator $\epsilon$ has a conformal weight $\Delta = -1$. As was shown in [1] the reparametrization modes can be used directly to compute the contribution of the stress-tensor block to the 4pt function $\langle XX\phi\phi \rangle$ upto $1/c$ order .

$$\begin{aligned}
\langle X_1 X_2 | T | \phi_3 \phi_4 \rangle &= \langle X_1 X_2 | T | T | \phi_3 \phi_4 \rangle \\
&= \mathbb{P}_{\alpha\beta}^{\rho\sigma} \mathbb{P}_{\mu\nu}^{\gamma\delta} \int d^d x \, d^d y \; \langle X_1 X_2 \, T^{\alpha\beta}(y) \rangle \, \langle \partial_\rho \epsilon_\sigma(y) \partial_\gamma \epsilon_\delta(x) \rangle \; \langle T^{\mu\nu}(x)\phi_3 \phi_4 \rangle \\
&= \Delta_\phi \Delta_X \langle \hat{\mathcal{B}}_{12}^{(1)} \hat{\mathcal{B}}_{34}^{(1)} \rangle \langle X_1 X_2 \rangle \langle \phi_3 \phi_4 \rangle
\end{aligned} \tag{2.6}$$

where in going from the first line to the second line we have used the fact that the projector squares to itself

$$|T| = |\widetilde{T}| = |T|^2 = \frac{k_{0,2}}{\pi^{d/2}c} \int d^d x \, \widetilde{T}(x)|0\rangle\langle 0|T(x) \tag{2.7}$$

In going from the second line to the third line in (2.6) we have used first the definition of $\mathbb{P}_{\mu\nu}^{\alpha\beta}$ in (2.5), then shift derivatives from $\epsilon$s to $T$s and use the conformal Ward identity for $T_{\mu\nu}$ insertions

$$\begin{aligned}
\langle \partial_\mu T_0^{\mu\nu} \phi_1 \phi_2 \rangle &= -\left[\delta^d(x_{01})\partial_1^\nu + \delta^d(x_{02})\partial_2^\nu\right]\langle \phi_1 \phi_2 \rangle + \frac{\Delta_\phi}{d}\partial_0^\nu\left[\delta^d(x_{01}) + \delta^d(x_{02})\right]\langle \phi_1 \phi_2 \rangle \\
\langle T_{0\mu}^\mu \phi_1 \phi_2 \rangle &= 0
\end{aligned} \tag{2.8}$$

to get rid of the integrals. Therefore we find the $\hat{\mathcal{B}}_{ij}^{(1)}$ to be[6]

$$\hat{\mathcal{B}}_{12}^{(1)} = \frac{1}{d}\left(\partial_\mu \epsilon_1^\mu + \partial_\mu \epsilon_2^\mu\right) - 2\frac{(\epsilon_1 - \epsilon_2)^\mu (x_{12})_\mu}{x_{12}^2}, \qquad x_{12}^\mu = (x_1 - x_2)^\mu \tag{2.9}$$

which are the bi-local bilinear operators constructed out of $\epsilon$s. These bi-locals have been used in a similar context in [50]. In 2d these take the form

$$\mathcal{B}_{12}^{(1)} = \partial_1 \epsilon_1 + \partial_2 \epsilon_2 - 2\frac{\epsilon_1 - \epsilon_2}{z_{12}}, \quad \bar{\mathcal{B}}_{12}^{(1)} = \bar{\partial}_1 \bar{\epsilon}_1 + \bar{\partial}_2 \bar{\epsilon}_2 - 2\frac{\bar{\epsilon}_1 - \bar{\epsilon}_2}{\bar{z}_{12}}, \qquad z_{12} = z_1 - z_2. \tag{2.10}$$

The 2pt functions for $\epsilon$s can be deduced from those of $\widetilde{T}$ which in turn can be readily obtained from general conformal covariance

$$\begin{aligned}
\langle \widetilde{T}_1^{\mu\nu} \widetilde{T}_2^{\alpha\beta} \rangle &= 2C_T \frac{k_{d,2}}{k_{d,0}} \mathbb{P}_{\rho\sigma}^{\mu\nu} \mathbb{P}_{\gamma\delta}^{\alpha\beta} \, I_{12}^{\rho\gamma} I_{12}^{\sigma\delta} \\
&\equiv -C_T \frac{k_{d,2}}{k_{d,0}} \mathbb{P}_{\rho\sigma}^{\mu\nu} \mathbb{P}_{\gamma\delta}^{\alpha\beta} \partial^\sigma \partial^\delta \left[I_{12}^{\rho\gamma} x_{12}^2 \log(\mu^2 x_{12}^2)\right]
\end{aligned} \tag{2.11}$$

Here, $\mu^2$ is introduced as an energy scale to make the argument dimensionless. This energy scale drops out upon taking the symmetric trace-less derivative and in the computation of correlations functions of physical operators.

---

[6]Our definition of $\hat{\mathcal{B}}_{ij}^{(1)}$ differs from that of [36] by a factor of $\Delta \stackrel{2d}{=} \frac{h+\bar{h}}{2}$.

## 2.2 Reparametrization modes in CFT$_2$

As we would be interested in holographic 2d CFTs in this paper we rewrite the above formalism in 2d. Using the left-right moving Euclidean co-ordinates $\{z, \bar{z}\}$ the metric and the inversion tensor takes the form

$$\eta_{ab} = \frac{1}{2}\begin{pmatrix} 0 & 1 \\ 1 & 0 \end{pmatrix}, \qquad I^{ab} = -2\begin{pmatrix} z/\bar{z} & 0 \\ 0 & \bar{z}/z \end{pmatrix} \tag{2.12}$$

Using which we can describe the shadow as of an operator $\mathcal{O}$ with conformal weights $\{h, \bar{h}\}$ as

$$\widetilde{\mathcal{O}}(z) = k_{\frac{h+\bar{h}}{2}, \frac{h-\bar{h}}{2}} \int d^2x \; \frac{O(x)}{(x-z)^{(2-2h)}(\bar{x}-\bar{z})^{(2-2\bar{h})}} \tag{2.13}$$

The shadow of the stress-tensor components are therefore given by the conformal integrals

$$\widetilde{T}(z) = \frac{2}{\pi}\int d^2x \; \frac{(z-x)^2}{(\bar{z}-\bar{x})^2} T(x), \quad \widetilde{\bar{T}}(\bar{z}) = \frac{2}{\pi}\int d^2x \; \frac{(\bar{z}-\bar{x})^2}{(z-x)^2} \bar{T}(\bar{x}) \tag{2.14}$$

In 2 dimensions the reparametrization modes defined in (2.5) take the form

$$\widetilde{T}^{zz} = \widetilde{T}(z) = \frac{c}{3}\bar{\partial}\epsilon, \qquad\qquad \widetilde{T}^{\bar{z}\bar{z}} = \widetilde{\bar{T}} = \frac{c}{3}\partial\bar{\epsilon} \tag{2.15}$$

Using the fact that $T = \widetilde{\widetilde{T}}$ one can show that[1, 36]

$$T_{zz} = T = -\frac{c}{12}\partial^3\epsilon, \qquad\qquad T_{\bar{z}\bar{z}} = \bar{T} = -\frac{c}{12}\bar{\partial}^3\bar{\epsilon} \tag{2.16}$$

Thus the stress-tensor itself can be seen as a descendent of the reparametrization mode $\epsilon$. We refer readers to section 3 of [1] for a discussion on effective action for the $\epsilon$ modes and for a d-dimensional generalization of the above relation[7]. The above relations impose strict consistency conditions which any correlation function involving $\epsilon$s have to satisfy:

$$\begin{aligned} \langle \partial^3\epsilon_1 \ldots \partial^3\epsilon_n \, \phi\phi \ldots\rangle &= \left(-\tfrac{12}{c}\right)^n \langle T_1 \ldots T_n \phi\phi \ldots\rangle, \\ \langle \bar{\partial}\epsilon_1 \ldots \bar{\partial}\epsilon_m \, \partial^3\epsilon_{m+1} \ldots \partial^3\epsilon_n \phi\phi \ldots\rangle &= \left(-\tfrac{3}{c}\right)^n 4^{n-m}\langle \widetilde{T}_1 \ldots \widetilde{T}_m T_{m+1} \ldots T_n \phi\phi \ldots\rangle, \end{aligned} \tag{2.17}$$

It is worth noting that the definition (2.15) doesn't readily imply holomorphic factorization. As we will see later in specific case of interest to us, the conditions of the form (2.17) will be used to fix the ambiguities arising from (2.15). These would give rise to a set of differential constraints. We will deal with these in the next section.

The $\widetilde{T}$ 2pt function in 2d is

$$\langle \widetilde{T}_1 \widetilde{T}_2 \rangle = \frac{2c}{3}\bar{\partial}_1\bar{\partial}_2 \left(z_{12}^2 \log(\mu^2 z_{12}\bar{z}_{12})\right) \tag{2.18}$$

We also note that

$$\langle T_i \widetilde{T}_j \rangle = \frac{c\pi}{3}\delta^{(2)}(x_i - x_j). \tag{2.19}$$

Therefore the 2pt function of the reparametrization modes consistent with all possible correlations of $T$ and $\widetilde{T}$ is given by

$$\langle \epsilon_1 \epsilon_2 \rangle = \frac{6}{c} z_{12}^2 \log(\mu^2 z_{12}\bar{z}_{12}). \tag{2.20}$$

Note that the above functional dependence is not unique upto addition of terms quatratic in distances but this would not effect computation of physically relevant quantities. The monodromy relations satisfied by

---

[7]In general dimensions the stress tensor is given by an differential operator action on the reparametrization modes, *c.f.* eq.3.12 of [1].

the conformal blocks allows one to factor out the contribution of the shadow blocks. In 2d this turns out to simply imposing holomorphic factorization [1].

$$\langle \epsilon_1 \epsilon_2 \rangle_{\text{phys}} = \frac{6}{c} z_{12}^2 \log(\mu^2 z_{12}). \tag{2.21}$$

The above propagator can then be used to compute the conformal block in (2.6)

$$\langle \mathcal{B}_{12}^{(1)} \mathcal{B}_{34}^{(1)} \rangle_{\text{phys}} = \frac{2}{c} \, z^2 \, {}_2F_1(2,2,4,z) \tag{2.22}$$

in terms of the the invariant cross ratio $z = \frac{z_{12} z_{34}}{z_{13} z_{24}}$. This is indeed the single graviton exchange correction computed to $1/c$ order for the 4pt function of pairs of light scalars [9]. As expected the global stress-tensor block in 2d captures only the level one states generated over the vacuum being exchanged.[8]

# 3   Ward identity for $\epsilon$s & 6pt block for the stress-tensor comb channel

The contribution of the stress-tensor block to any correlation function can be thought of as being obtained by inserting the stress-tensor projector $|T\rangle$ as defined in (2.6). Every such insertion can be thought of as introducing a power of $1/c$. The stress-tensor legs propagating within the diagram can all be decomposed into a web of legs connected with 3pt vertices of stress-tensor and its shadow. Consequently fusing any three of the stress-tensors introduces a power of $c$ (*i.e.* $\langle T_1 T_2 T_3 \rangle \sim c$)[9]. Given that insertion of the stress-tensor projectors $|T\rangle$ can be recast in terms of insertion of $\epsilon$s as in (2.6) we must be able to express the stress-tensor block contribution as expectation values involving insertions of epsilons. Insertions of stress-tensors in 2d is completely fixed in terms of the 2d Ward identity. We would therefore like to understand what this implies for the $\epsilon$ insertions.

In this section we review in slightly different way the Ward identity associated with the insertion of a single reparametrization mode into a 2pt function of primaries [1]. We then proceed in subsection-3.2 to analyse the Ward identity for 2 insertions of the reparametriaztion mode in a 2pt function in CFT$_2$. We use this expression in subsection-3.3 to compute the Virasoro contribution to the 6pt stress-tensor comb channel consisting of 3 pair-wise identical primaries. This has not been computed in the literature yet and is one of the central results of this paper. We also show here that the method used to obtain the 4pt function of 2 pair-wise identical operators in subsection-3.1 can be generalised to obtain the vacuum block of 6pt function of 2 pairs of identical triplets *i.e.* $\langle \phi_{1,2,3} \psi_{4,5,6} \rangle$. Similarly the 6pt comb channel expressions of 3 identical pairs can be generalized to obtain the 9pt comb channel expressions for 3 identical triplets *i.e.* $\langle X_{3,5,7} \phi_{0,1,2} Y_{4,6,8} \rangle_{T,comb}^{Vir}$. This form of generalization is made possible specifically by the use of the reparametrization mode Ward identities used to obtain the result.

## 3.1   Single reparametrization mode Ward identity

The expression we derive here for the Ward identity of a single insertion of the reparametrization mode is applicable in CFT$_d$ [1]. Therefore only in this subsection we give expressions that are valid in any $d$ and also write 2d expressions in the tensorial notation applicable in any $d$. We first begin with $\epsilon$ insertions in 2-pt. functions of scalars $\phi$ of conformal weight $\{h,h\}$ *i.e.* $\Delta = 2h$. This can be obtained from

$$\langle \widetilde{T}_{-1}^{ab} \phi_1 \phi_2 \rangle = \frac{2}{\pi} \int_0 \frac{I_{-10}^{a\bar{a}} I_{-10}^{b\bar{b}} \langle T_0 \phi_1 \phi_2 \rangle_{\bar{a}\bar{b}}}{X_{-10}^0} \tag{3.1}$$

---

[8]$L_{-1}^n |T\rangle \sim L_{-n}\rangle$ since $T\rangle = L_{-2}\rangle$.

[9]In 2d we have $\langle T_1 T_2 T_3 \rangle = -c \frac{1}{z_{12}^2 z_{23}^2 z_{13}^2}$, $\langle \widetilde{T}_1 \widetilde{T}_2 \widetilde{T}_3 \rangle = c \frac{8}{3} \frac{z_{12} z_{23} z_{13}}{\bar{z}_{12} \bar{z}_{23} \bar{z}_{13}}$  $\langle T_1 \widetilde{T}_2 \widetilde{T}_3 \rangle = c \frac{8}{3} \frac{z_{23}^4}{z_{12}^2 \bar{z}_{23}^2 z_{13}}$
$\langle T_1 T_2 \widetilde{T}_3 \rangle = c \frac{\bar{z}_{12} \bar{z}_{23} z_{13}}{z_{12}^5 \bar{z}_{23} \bar{z}_{13}}$

where $T_{zz} = T$, $\widetilde{T}^{zz} = \widetilde{T}$. Where we denote compactly the conformal integral as

$$\int_i \equiv \int d^2 X_i. \tag{3.2}$$

This can be evaluated by evaluating $\langle \widetilde{K}^{ab}_{(1)} \phi_{(2)} \phi_{(3)} \rangle$ where $K_{ab}$ is spin 2 primary with weight $\delta$ and then taking the limit $\delta \to d = 2$[10]. Alternatively, this should match the expected answer for

$$\langle \widetilde{T}^{ab}_{-1} \phi_1 \phi_2 \rangle = \left. \frac{2h}{\pi} \frac{I^{a\bar{a}}_{-11} I^{12}_{\bar{a}\bar{b}} I^{b\bar{b}}_{-12}}{(X_{-11} X_{-12})^{\delta/2} X_{12}^{2h-\delta/2}} \right|_{\delta \to 0} \tag{3.3}$$

We can insert a $|T|$ in between the $\widetilde{T}$ and $\phi$s in the $rhs$ above as any other insertion (including $I$) would yield 0. Here we assume that $\langle \widetilde{T}^{ab} \rangle = 0$.

$$\begin{aligned}
\langle \widetilde{T}^{\mu\nu}_{-1} | T | \phi_1 \phi_2 \rangle &= \mathbb{P}^{\rho\sigma}_{\alpha\beta} \int d^d x \langle \widetilde{T}^{\mu\nu} \partial_\rho \epsilon_\sigma(x) \rangle \langle T^{\alpha\beta}(x) \phi_1 \phi_2 \rangle \\
&= \frac{c\pi^{d/2}}{k_{0,2}} \mathbb{P}^{\rho\sigma}_{\alpha\beta} \mathbb{P}^{\mu\nu}_{\gamma\delta} \int d^d x \langle \partial^\gamma \epsilon^\delta_{-1} \partial_\rho \epsilon_\sigma(x) \rangle \langle T^{\alpha\beta}(x) \phi_1 \phi_2 \rangle \\
&= \frac{c\pi^{d/2}}{k_{0,2}} \mathbb{P}^{\rho\sigma}_{\alpha\beta} \langle \partial^\gamma \epsilon^\sigma_{-1} \hat{\mathcal{B}}^{(1)}_{12} \rangle \langle \phi_1 \phi_2 \rangle.
\end{aligned} \tag{3.4}$$

where as before we used the (2.5) & (2.7) and shift derivatives onto $T^{\alpha\beta}(x)$ above then use the Ward identity (2.8) as before. This in 2d yields[11]

$$\langle \widetilde{T}_{-1} \phi_1 \phi_2 \rangle = \frac{ch}{3} \bar{\partial}_{-1} \langle \epsilon_{-1} \mathcal{B}^{(1)}_{12} \rangle \langle \phi_1 \phi_2 \rangle \tag{3.5}$$

Therefore we find

$$\langle \epsilon_{-1} \phi_1 \phi_2 \rangle = \langle \epsilon_{-1} \mathcal{B}^{(1)}_{12} \rangle \langle \phi_1 \phi_2 \rangle \tag{3.6}$$

This relation (along with its barred counter-part) had already found it's use in computing the 4pt stress-tensor block [1] as follows

$$\begin{aligned}
\langle \phi_1 \phi_2 | T | \phi_3 \phi_4 \rangle &= -\mathbb{P}^{\alpha\beta}_{\mu\nu} \int_{-1} \langle \phi_1 \phi_2 \, \partial_\beta \epsilon_{-1\alpha} \rangle \langle T^{\mu\nu}_{-1} \phi_3 \phi_4 \rangle \\
&= -\Delta \mathbb{P}^{\alpha\beta}_{\mu\nu} \int_{-1} \langle \phi_1 \phi_2 \rangle \langle \hat{\mathcal{B}}^{(1)}_{12} \partial_\beta \epsilon_{-1\alpha} \rangle \langle T^{\mu\nu}_{-1} \phi_3 \phi_4 \rangle \\
&= \Delta^2 \langle \hat{\mathcal{B}}^{(1)}_{12} \hat{\mathcal{B}}^{(1)}_{34} \rangle \langle \phi_1 \phi_2 \rangle \langle \phi_3 \phi_4 \rangle
\end{aligned} \tag{3.7}$$

where we use definition of $\epsilon$s (2.5),(2.7) and the Ward identity in the going from the second to the third line along with $\epsilon^z = \epsilon$, $\epsilon^{\bar{z}} = \bar{\epsilon}$. Note that in the last line above $\hat{\mathcal{B}}^{(1)}_{ij}$ consists of sum of holomorphic and anti-holomorphic parts while that in (3.6) consists of only the holomorphic sector[12]. The above expression was computed in [1] by a slightly different approach of squaring the projectors and using the stress-tensor Ward-identities. In 2d the above expression for $\langle \hat{\mathcal{B}}^{(1)}_{12} \hat{\mathcal{B}}^{(1)}_{34} \rangle$ is given by (2.22).

## 3.2 Double reparametrization mode Ward identity

From now on we restrict to CFT$_2$ in this paper. We would next like to compute $\langle \phi_1 \phi_2 \epsilon_3 \epsilon_4 \rangle$. To this end we turn to compute $\langle \widetilde{T}_{-1} \widetilde{T}_0 \phi_1 \phi_2 \rangle$, here we do not have the benefit of the general structure being fixed by global conformal invariance as was in the previous case. It would be instructive to write down the full Ward identity

$$\langle T_{-1} T_0 \phi_1 \phi_2 \rangle = \frac{c/2}{z_{-10}^4} \langle \phi_1 \phi_2 \rangle + \sum_{i=0}^{2} \left( \frac{h_i}{z_{-1i}^2} + \frac{\partial_i}{z_{-1i}} \right) \frac{h z_{12}^2}{z_{01}^2 z_{02}^2} \frac{1}{(z_{12} \bar{z}_{12})^{2h}} \tag{3.8}$$

---

[10]In other words $\langle \widetilde{T}^{ab}_{-1} \phi_1 \phi_2 \rangle$ is also fixed by global conformal invariance.

[11]Note that in 2d the use of $|T|$ as an integral yields the projector onto only the global states of the stress-tensor, however this is enough to fix the structure of 3pt functions as they are completely determined by only global symmetries.

[12]Any insertion of $|T|$ in 2d would consist of a holomorphic term and an anti-holomorphic term, (3.6) deals with only the holomorphic sector.

where $\langle\phi_1\phi_2\rangle = (z_{12}\bar{z}_{12})^{-2h} \equiv X_{12}^{-2h}$, $h_0 = 2$ and $h_{1,2} = h$. we would like to evaluate the shadow of the above *rhs wrt* coordinates $X_{-1}\&X_0$. The shadow of the first term is simply the obtained by evaluating $\langle\tilde{B}_3^{ab}\tilde{B}_4^{cd}\rangle$ and taking the conformal dimension of $\tilde{B}$ to zero. The second term as a whole is globally conformally invariant but not in parts. It turns out one can split it into 3 parts each of which are globally conformally invariant[13]

$$\langle T_{-1}T_0\phi_1\phi_2\rangle = \frac{c/2}{z_{-10}^4}\langle\phi_1\phi_2\rangle + \left(\frac{(h-1)z_{12}^2}{z_{-11}^2 z_{-12}^2} + \frac{z_{01}^2}{z_{-10}^2 z_{-11}^2} + \frac{z_{02}^2}{z_{-10}^2 z_{-12}^2}\right)\frac{hz_{12}^2}{z_{01}^2 z_{02}^2}\frac{1}{(z_{12}\bar{z}_{12})^{2h}}. \tag{3.9}$$

The benefit of expressing the Ward identity in conformally invariant terms is that we can make use of the expression

$$\langle T_0\phi_1\phi_2\rangle = -\frac{hz_{12}^2}{z_{01}^2 z_{02}^2}\langle\phi_1\phi_2\rangle \implies \langle\widetilde{T}_4\phi_1\phi_2\rangle = \frac{c}{3}\langle\bar{\partial}\epsilon_4\phi_1\phi_2\rangle = \frac{2}{\pi}\int_0 \frac{z_{40}^2}{\bar{z}_{40}^2}\frac{hz_{12}^2}{z_{01}^2 z_{02}^2}\langle\phi_1\phi_2\rangle \tag{3.10}$$

using which we can recast the Ward identity (3.9) as

$$\langle T_{-1}T_0\phi_1\phi_2\rangle = \langle T_{-1}T_0\rangle\langle\phi_1\phi_2\rangle + h(h-1)\langle T_{-1}\mathcal{B}_{12}^{(1)}\rangle\langle T_0\mathcal{B}_{12}^{(1)}\rangle\langle\phi_1\phi_2\rangle$$

$$+h\langle T_0\mathcal{B}_{12}^{(1)}\rangle\left(\langle T_{-1}\mathcal{B}_{01}^{(1)}\rangle + \langle T_{-1}\mathcal{B}_{02}^{(1)}\rangle\right)\langle\phi_1\phi_2\rangle \tag{3.11}$$

Although not explicitly manifest the last term is symmetric under $0 \leftrightarrow -1$. Therefore we can write a Ward identity for the shadow stress-tensor $\widetilde{T}$ as

$$\langle\widetilde{T}_3\widetilde{T}_4\phi_1\phi_2\rangle = \langle\widetilde{T}_3\widetilde{T}_4\rangle\langle\phi_1\phi_2\rangle + h(h-1)\langle\widetilde{T}_3\mathcal{B}_{12}^{(1)}\rangle\langle\widetilde{T}_4\mathcal{B}_{12}^{(1)}\rangle\langle\phi_1\phi_2\rangle$$

$$+\frac{2h}{\pi}\int_0 \frac{z_{40}^2}{\bar{z}_{40}^2}\langle T_0\mathcal{B}_{12}^{(1)}\rangle\left(\langle\widetilde{T}_3\mathcal{B}_{01}^{(1)}\rangle + \langle\widetilde{T}_3\mathcal{B}_{02}^{(1)}\rangle\right)\langle\phi_1\phi_2\rangle \tag{3.12}$$

where we have used

$$\frac{\langle T_0\phi_1\phi_2\rangle}{\langle\phi_1\phi_2\rangle} = h\langle T_0\mathcal{B}_{12}^{(1)}\rangle, \qquad \frac{\langle\widetilde{T}_4\phi_1\phi_2\rangle}{\langle\phi_1\phi_2\rangle} = h\langle\widetilde{T}_4\mathcal{B}_{12}^{(1)}\rangle \tag{3.13}$$

which can be verified given the basic definitions in the previous section. Making use of the definition (2.15) we can easily write the corresponding Ward identity for the reparametrization modes, except for the last term above. Restricting further to only the physical block by taking only the holomorphic sector above yields

$$\frac{\langle\epsilon_3\epsilon_4\phi_1\phi_2\rangle_{\text{phys}}}{\langle\phi_1\phi_2\rangle} = \langle\epsilon_3\epsilon_4\rangle_{\text{phys}} + h(h-1)\langle\epsilon_3\mathcal{B}_{12}^{(1)}\rangle_{\text{phys}}\langle\epsilon_4\mathcal{B}_{12}^{(1)}\rangle_{\text{phys}} + h\left(\frac{12}{c}\right)^2\mathcal{C}_{\text{phys}}^{(2)}$$

$$\tag{3.14}$$

where the last term $\mathcal{C}^{(2)}$ needs to be determined. We note here that this term can be determined in 2 possible ways: ($i$) by explicitly solving the integral in the last term in (3.12) by making use of the integrals listed in Appendix B, and then writing the result as total derivatives of $\bar{z}_3$ & $\bar{z}_4$. or ($ii$) by solving consistency conditions of the type (2.17) some of which result in solving differential equations in the cross ratio. Method ($i$) is actually insufficient for getting the right answer as there can be ambiguities in adding a term which vanish upon differentiation *wrt* $\bar{z}_{3,4}$. Upon integration of the last term in (3.12) (as done in Appendix C) one can write $\mathcal{C}^{(2)}$ as

$$\mathcal{C}_{\text{phys}}^{(2)} = \langle\epsilon_3\epsilon_4\rangle_{\text{phys}}\left[4 + \left(-2 + \frac{4}{z}\right)\log(1-z)\right] + z_{34}^2\mathcal{F}(z). \tag{3.15}$$

where $z = \frac{z_{12}z_{34}}{z_{13}z_{24}}$; we simply write the resultant integral as a total derivative of $\bar{z}_{3,4}$. Here we have added an extra term $z_{34}^2\mathcal{F}(z)$ which would be required to make (3.14) satisfy the the constraints (2.17).

---

[13]We see this by counting powers of variables to be integrated *i.e.* $X_{-1}\&X_0$, and they must add up to $-d = -2$ for each of them.

Method ($ii$) solves for constrains due to the first of the relation in (2.15) implying that we must get the Ward identity (3.8) (appropriately normalized) upon using this relation on each of the $\epsilon$s in (3.14).

$$\langle \partial^3 \epsilon_3 \partial^3 \epsilon_4 \phi_1 \phi_2 \rangle = \left( \frac{12}{c} \right)^2 \langle T_3 T_4 \phi_1 \phi_2 \rangle \tag{3.16}$$

This consistency condition is satisfied term by term and the first 2 terms in (3.14) satisfy this. Apart from the above condition we further have to satisfy

$$\langle \partial^3 \epsilon_3 \bar{\partial} \epsilon_4 \phi_1 \phi_2 \rangle = -\frac{36}{c^2} \langle T_3 \widetilde{T}_4 \phi_1 \phi_2 \rangle \tag{3.17}$$

but this condition constrains pieces that give contribution to the shadow block in (3.14)[14]. To see this we note that the Ward identity (3.11) implies

$$\langle T_3 \widetilde{T}_4 \phi_1 \phi_2 \rangle = \langle T_3 \widetilde{T}_4 \rangle \langle \phi_1 \phi_2 \rangle + \tfrac{(h-1)}{h} \langle T_3 \mathcal{B}_{12}^{(1)} \rangle \langle \widetilde{T}_4 \mathcal{B}_{12}^{(1)} \rangle \langle \phi_1 \phi_2 \rangle + \langle T_3 \mathcal{B}_{12}^{(1)} \rangle \left( \langle \widetilde{T}_4 \mathcal{B}_{31}^{(1)} \rangle + \langle \widetilde{T}_4 \mathcal{B}_{32}^{(1)} \rangle \right) \langle \phi_1 \phi_2 \rangle \tag{3.18}$$

Making use of the fact

$$\langle T_3 \phi_1 \phi_2 \rangle = -\frac{h \, z_{12}^2}{z_{23}^2 z_{13}^2} \langle \phi_1 \phi_2 \rangle, \qquad \langle \widetilde{T}_4 \phi_1 \phi_2 \rangle = -4 \frac{h \, z_{41} \bar{z}_{12} z_{24}}{\bar{z}_{41} z_{12} \bar{z}_{24}} \langle \phi_1 \phi_2 \rangle = \langle \widetilde{T}_4 \mathcal{B}_{12}^{(1)} \rangle \langle \phi_1 \phi_2 \rangle \tag{3.19}$$

and noting that expressing $\langle \widetilde{T}_4 \phi_1 \phi_2 \rangle$ as $\bar{\partial}_4 \langle \epsilon_4 \phi_1 \phi_2 \rangle$ implies that only $\langle \epsilon_4 \phi_1 \phi_2 \rangle_{\text{shdw}}$ contributes to $\langle \widetilde{T}_4 \phi_1 \phi_2 \rangle$. This is merely because $\langle \epsilon_4 \phi_1 \phi_2 \rangle_{\text{phys}} \sim \langle \epsilon_4 \mathcal{B}_{ij}^{(1)} \rangle_{\text{phys}} \langle \phi_1 \phi_2 \rangle$ does not contain inverse powers of $z_4$[15].

The terms in the *rhs* of (3.14) are directly related to the terms in the Ward identity (3.9). The constraint (3.16) is satisfied by all but the last term in (3.14) which is related to the last 2 terms in (3.9). Therefore the consistency condition satisfied by the physical part of $\mathcal{C}^{(2)}$ is

$$\partial_3^3 \partial_4^3 \mathcal{C}_{\text{phys}}^{(2)} = \frac{z^2 (2 - 2z + z^2)}{z_{34}^4 (z-1)^2}. \tag{3.20}$$

It turns out that $\mathcal{C}_{\text{phys}}^{(2)}$ can be determined to be

$$\mathcal{C}_{\text{phys}}^{(2)} = z_{34}^2 \mathcal{A}(z)$$

$$\mathcal{A}(z) = \frac{1}{16z^2} \Big[ \left( 4z^2 + 2z - 5 \right) \text{Li}_2 \left( \tfrac{1}{1-z} \right) + \left( z^2 + 8z - 5 \right) \text{Li}_2(1 - z) + 5 \left( z^2 - 1 \right) \text{Li}_2(z)$$

$$+ 5(2z - 1) \text{Li}_2 \left( \tfrac{z}{z-1} \right) - 2 \left( z^2 - 6z + 6 \right) \log^2(1 - z) + 8z^2 (\log(z - 1) - 2\log(z))$$

$$+ 5 \log(1 - z)((2z - 1)\log(z - 1) + (z - 2)z \log(z))\Big] \tag{3.21}$$

$\mathcal{A}(z)$ is explicitly symmetric under $(1 \leftrightarrow 2)$ & $(3 \leftrightarrow 4)$. Of course $\mathcal{A}$ is not uniquely determined but the ambiguities- which can be made explicit in the process of finding $\mathcal{A}$, do not contribute to anything physical. For example adding a function of the type

$$\frac{z_{34}^2}{z^2} \left( a_1 z^2 + a_2 z + a_3 + (b_1 z^2 + b_2 z + b_3) \log(1 - z) \right) \tag{3.22}$$

to $\mathcal{A}(z)$ also satisfies the same consistency condition[16]. Note one can similarly satisfy the constraint (3.17) too, but as mentioned before this would only give contributions to the shadow part in (3.14). With the above

---

[14] This requires having a term proportional to $\log \bar{z}$.

[15] Unlike the constraint for $\langle \epsilon_1 \epsilon_2 \rangle$ where the constraint $\langle T_1 \widetilde{T}_2 \rangle = -\frac{3\pi}{c} \delta_{12}^{(2)} = \frac{3}{c} \bar{\partial}_2 \frac{1}{z_{12}}$ does constrain $\langle T_1 \epsilon_2 \rangle$.

[16] One can explicitly show that adding the above function (3.22) to $\mathcal{A}$ doesn't effect the answer we would eventually compute in (3.27).

expression for $\mathcal{C}^{(2)}_{\mathrm{phys}}$ we have completely solved for the reparametrization mode Ward identity (3.14) with 2 $\epsilon$ insertions in a 2pt function. We will see in the next subsection how this would be used to compute the contribution of the stress tensor block to higher point correlators.

It must be noted that although $\langle \epsilon\epsilon \rangle$ themselves are not conformally invariant- $\epsilon$s transforming like an operator with conformal dimension $-1$; the 2pt functions of the bilinears constructed out of $\epsilon$s however are conformally invariant

$$\frac{\langle \phi_1\phi_2\phi_3\phi_4 \rangle}{\langle \phi_1\phi_2 \rangle \langle \phi_3\phi_4 \rangle} = h_\phi^2 \langle \mathcal{B}^{(1)}_{12}\mathcal{B}^{(1)}_{34} \rangle. \tag{3.23}$$

One can also foresee that any computation of a block would involve $\mathcal{B}^{(1)}_{ij}$- as in the case of the 4pt stress-tensor block; and not $\epsilon_i$s themselves. Therefore we consider the insertions of the reparametrizations mode operator $\epsilon$s in form of bilocals $\mathcal{B}_{ij}$ inside the 2pt function $\langle \phi_1\phi_2 \rangle$ i.e.

$$\frac{\left\langle \mathcal{B}^{(1)}_{35}\mathcal{B}^{(1)}_{46}\phi_1\phi_2 \right\rangle}{\langle \phi_1\phi_2 \rangle}. \tag{3.24}$$

This can be readily seen as conformally invariant as it is obtained from a particular diagram contributing the global conformal block of 6-pt function of pair-wise equal operators:

$$\frac{\langle X_3 X_5 \, |T|_g \, \phi_1\phi_2 \, |T|_g \, Y_4 Y_6 \rangle}{\langle \phi_1\phi_2 \rangle \langle X_3 X_5 \rangle \langle Y_4 Y_6 \rangle} = \left( \frac{3}{\pi c} \right)^2 \int_{3',4'} \frac{\langle \widetilde{T}_{3'}\widetilde{T}_{4'}\phi_1\phi_2 \rangle \langle T_{3'}X_3 X_5 \rangle \langle T_{4'}Y_4 Y_6 \rangle}{\langle \phi_1\phi_2 \rangle \langle X_3 X_5 \rangle \langle Y_4 Y_6 \rangle}$$

$$= h_X h_Y \frac{\left\langle \mathcal{B}^{(1)}_{35}\mathcal{B}^{(1)}_{46}\phi_1\phi_2 \right\rangle}{\langle \phi_1\phi_2 \rangle} \tag{3.25}$$

where we use the sub-script $g$ in $|T|_g$ to denote projection onto global states associated with the stress-tensor. (From now on we use $|\mathcal{O}|$ to denote the full Virasoro block while $|\mathcal{O}|_g$ to denote just the global block associated with any operator $\mathcal{O}$.) This consists of a connected diagram[17] contributing to the vacuum block of the 6-pt pairwise equal operators and it only need be normalized by the 2-pt functions[18]. Using the definition of the $\epsilon$ as derivative of $\widetilde{T}$ and then the Ward identity after integrating by parts as before, one is left with (3.24) upto proportionality constants which depend on the operator dimensions. Using (3.14) we can expand (3.24) as

$$\frac{\left\langle \mathcal{B}^{(1)}_{35}\mathcal{B}^{(1)}_{46}\phi_1\phi_2 \right\rangle}{\langle \phi_1\phi_2 \rangle} = \langle \mathcal{B}^{(1)}_{35}\mathcal{B}^{(1)}_{46} \rangle + h_\phi(h_\phi - 1)\langle \mathcal{B}^{(1)}_{35}\mathcal{B}^{(1)}_{12} \rangle \langle \mathcal{B}^{(1)}_{12}\mathcal{B}^{(1)}_{46} \rangle + h \left( \frac{12}{c} \right)^2 \mathcal{K}^{(2)} \tag{3.26}$$

where $\mathcal{K}^{(2)}$ captures the contribution of the last term in (3.14). The first 2 terms are built of the familiar vacuum 4-pt conformal blocks and their contribution to the different OTOs can be therefore readily discerned. $\mathcal{K}^{(2)}$ an be written as[19]

$$\mathcal{K}^{(2)} = \left[ \partial_3\partial_4 - 2 \left( \frac{\partial_3}{z_{46}} + \frac{\partial_4}{z_{35}} \right) + \frac{4}{z_{35}z_{46}} \right] \mathcal{C}^{(2)} + (3 \leftrightarrow 5) + (4 \leftrightarrow 6) + (3 \leftrightarrow 5, 4 \leftrightarrow 6) \tag{3.27}$$

We do not give the explicit expression for $\mathcal{K}^{(2)}$ as it would be too cumbersome. However we would choose to extract the behaviours of those functions which posses branch cuts in the conformally invariant cross ratios which we do in the next section. Before doing so we would like to understand what exactly would we be computing as in the 4pt case.

---

[17]This expression does contain disconnected pieces, refer to subsection(3.3) for the discussion.

[18]We would have to explicitly remove the contribution coming from the first term in the Ward identity as this would correspond to the fusion of the stress-tensors in the projectors and would be of a lower order in $1/c$. We will see this explicitly in what follows.

[19]The relation between $\mathcal{C}^{(2)}$ and $\mathcal{K}^{(2)}$ can be obtained by expanding $\langle \mathcal{B}^{(1)}_{35}\mathcal{B}^{(1)}_{46}\phi_1\phi_2 \rangle'$ using (2.10).

### 3.2.1 Non-linear contributions in star channel *via* holography

We take a small detour to note how the vacuum block of the 6pt function in the star channel was computed using the reparametrization mode $\epsilon$ in [36] by Anous & Haehl. The authors made use of a non-linear generalization of $\mathcal{B}_{ij}^{(1)}$- denoted as $\mathcal{B}_{ij}$, inspired by the work of Cotler & Jensen in [22]. In [22] the authors derived an effective action for CFT$_2$ stress-tensor given in terms of $\mathcal{B}_{ij}$. In terms of which the connected contribution to the 6pt vacuum block is given as

$$\mathcal{V}_T^{(6)} = \left. \frac{\langle \mathcal{B}_{12}\mathcal{B}_{35}\mathcal{B}_{46}\rangle \langle \mathcal{B}_{12}\rangle \langle \mathcal{B}_{46}\rangle \langle \mathcal{B}_{35}\rangle}{\langle \mathcal{B}_{12}\mathcal{B}_{46}\rangle \langle \mathcal{B}_{12}\mathcal{B}_{35}\rangle \langle \mathcal{B}_{35}\mathcal{B}_{46}\rangle} \right|_{\text{phys}} \tag{3.28}$$

Their form is deduced by generalizing the conformal transformation of 2pt functions of primaries to arbitrary co-ordinate reparametrizations. $\mathcal{B}_{ij}$ is then defined as the ratio of 2pt functions of primaries in different frames.

$$z \to f(z, \bar{z}) = z + \epsilon(z, \bar{z}) + \mathcal{O}(\epsilon^2) \tag{3.29}$$

$$\mathcal{B}_{12} \approx \mathcal{B}_{h,12} = z_{12}^{2h} \left( \frac{\partial f(z_1, \bar{z}_1)\partial f(z_2, \bar{z}_2)}{(f(z_1, \bar{z}_1) - f(z_2, \bar{z}_2))^2} \right)^h = 1 + \sum_{p\geq 1} \mathcal{B}_{12}^{(p)} \tag{3.30}$$

$$\implies \mathcal{B}_{12}^{(1)} = h\left[ \partial\epsilon_1 + \partial\epsilon_2 - 2\frac{(\epsilon_1 - \epsilon_2)}{z_{12}} \right] \tag{3.31}$$

This allows the authors of [36] to have the first sub-leading correction ($\mathcal{O}(c^{-2})$ in this case) to come from truly connected 6pt diagram. It is important to note that (3.29) is not holomorphic and the effective action for stress-tensor propagation as derived in [22] is obtained from the gravitational path-integral in $AdS_3$. It is therefore plausible to expect that a formalism to compute higher point vacuum blocks must exist utilising the reparametrization modes $\epsilon$s but without recourse to holography.

### 3.3 Channel diagrammatics

We would next like to understand to what kind of correlation functions does the above expectation value (3.26) give an answer to. We note the expression for the global 6pt stress tensor comb channel written using the shadow formalism from [49]

$$\frac{\langle X_3 X_5 |T| \phi_1 |\phi_g| \phi_2 |T| Y_4 Y_6\rangle}{\langle \phi_1\phi_2\rangle\langle X_3 X_5\rangle\langle Y_4 Y_6\rangle} = \frac{\Gamma(2h_\phi)}{\pi(1 - 2\bar{h}_\phi)} \left( \frac{3}{\pi c} \right)^2 \int_{3',4',5'} \frac{\langle X_3 X_5 T_{3'}\rangle\langle \widetilde{T}_{3'}\phi_1\,\phi_{5'}\rangle\langle \widetilde{\phi}_{5'}\phi_2\widetilde{T}_{4'}\rangle\langle T_{4'} Y_4 Y_6\rangle}{\langle \phi_1\phi_2\rangle\langle X_3 X_5\rangle\langle Y_4 Y_6\rangle} \tag{3.32}$$

where we have inserted an projector $|\phi_g|$ which is a 2d analogue of (2.3). The suffix indicates that this projector in 2d only projects onto global states associated with the primary operator $\phi$. Figure 1(a). *Note, demanding that the $X$ & $Y$ pairs fuse to give the stress-tensor fixes the operator propagating in the internal leg.*

Allowing all possible states to propagate between $\phi_1$ and $\phi_2$ in the above correlator implies considering the Virasoro descendants of $\phi$, this amounts to not inserting any projector between $\phi_1$ and $\phi_2$. This can be seen as a computation of the Virasoro contribution to the 6pt stress-tensor Comb Channel as (3.26) gives a contribution to

$$\frac{\langle X_3 X_5 |T|_g \phi_1\phi_2 |T|_g Y_4 Y_6\rangle}{\langle \phi_1\phi_2\rangle\langle X_3 X_5\rangle\langle Y_4 Y_6\rangle} = \left( \frac{3}{\pi c} \right)^2 \int_{3',4'} \frac{\langle \widetilde{T}_{3'}\widetilde{T}_{4'}\phi_1\phi_2\rangle\langle T_{3'} X_3 X_5\rangle\langle T_{4'} Y_4 Y_6\rangle}{\langle \phi_1\phi_2\rangle\langle X_3 X_5\rangle\langle Y_4 Y_6\rangle}; \tag{3.33}$$

We can see that the above expression is the same as the Comb channel Virasoro block evaluated upto $\mathcal{O}(1/c^2)$. One can alternatively use the traditional projectors built out of Virasoro generators to see this too. We write

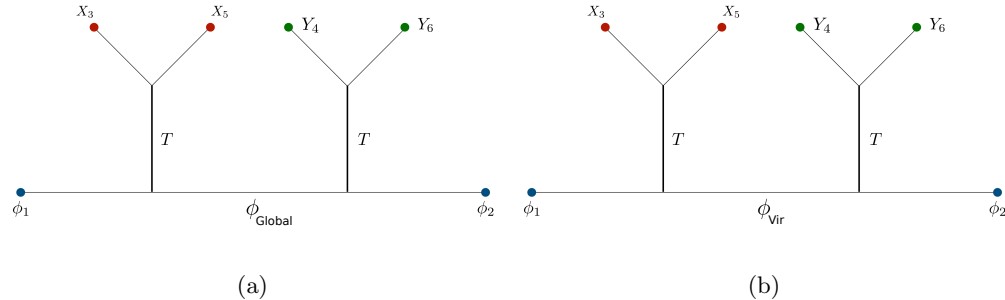

(a)                  (b)

**Figure 1**: (a) Global part of the stress-tensor comb channel. (b)Virasoro block of the stress-tensor comb channel. Here as compared to the global comb channel stress tensor block (a) all descendents of the $\phi$ operator are allowed to propagate.

the expression for the comb channel in Figure:1(b) upto $\mathcal{O}(1/c^2)$ as

$$\widetilde{V}_{T,\text{comb}}^{(6)} = \sum_{i,j} \frac{\langle X_3 X_5 L_{-i}\rangle\langle L_i\,\phi_1\phi_2\,L_{-j}\rangle\langle L_j\,Y_4 Y_6\rangle}{\langle\phi_1\phi_2\rangle\langle X_3 X_5\rangle\langle Y_4 Y_6\rangle\langle L_i L_{-i}\rangle\langle L_j L_{-j}\rangle} \tag{3.34}$$

where we do not insert any projector between the operators $\phi_1$ and $\phi_2$. As was evident from the computation of the 4pt. vacuum block we can replace the vacuum block projector at $\mathcal{O}(1/c)$ with the global stress-tensor projector

$$I_1 = \sum_i \frac{L_{-i}\rangle\langle L_i}{\langle L_i L_{-i}\rangle} \equiv \int dx_0\ T_0\rangle\langle\widetilde{T}_0 \tag{3.35}$$

thus the expressions (3.34) and (3.33) are identical. Therefore (3.33) computes a contribution to the Virasoro Comb channel block for Figure1 upto $\mathcal{O}(1/c^2)$.

$$\widetilde{V}_{T,\text{comb}}^{(6)\text{Vir}} = h_X h_Y \frac{\langle\mathcal{B}_{35}^{(1)}\mathcal{B}_{46}^{(1)}\phi_1\phi_2\rangle'}{\langle\phi_1\phi_2\rangle} \tag{3.36}$$

where the prime implies that we retain only the $1/c^2$ terms in (3.26) *i.e.* we ignore the first term in (3.26) as it contributes to a disconnected diagram and is of order $\mathcal{O}(1/c)$. Therefore the reparametrization mode Ward identity of the form $\langle\epsilon\epsilon\phi\phi\rangle$ enables us to compute the Virasoro contribution to the stress-tensor comb channel of 3 pairs of identical operators.

There are further contributions to the comb channel at $\mathcal{O}(1/c^2)$ order which can be seen as follows. In obtaining the above answer we make use of the identification (3.35). Consider the expression

$$V_{T,comb}^{(6)Vir} = \frac{\langle X_3 X_5\,|1 + I_1 + I_2 + \dots|\,\phi_1\,\phi_2\,|1 + I_1 + I_2 + \dots|\,Y_4 Y_6\rangle}{\langle\phi_1\phi_2\rangle\langle X_3 X_5\rangle\langle Y_4 Y_6\rangle} \tag{3.37}$$

where $I_n$ is the projector onto the vacuum at order $1/c^n$ and the $\dots$ denote higher order terms in $1/c$ expansion. $V_{T,comb}^{(6)Vir}$ can be regarded as the vacuum 6pt comb channel block. It is obvious that connected diagrams begin to start contributing at $\mathcal{O}(1/c^2)$. To obtain the full contribution at this order we would have to consider contributions upto $I_2$. Writing the projectors explicitly we have

$$\mathbb{I} = 1 + I_1 + I_2 + \dots$$

$$I_1 = \sum_i \mathcal{N}_{i,i}^{-1}\ L_{-i}\rangle\langle L_i$$

$$I_2 = \sum_{m,n} L_{-(m,n)}\rangle \left[ \mathcal{N}^{-1}_{(m,n),(m,n)}\langle L_{(m,n)} + \mathcal{N}^{-1}_{(m,n),(m+n)}\langle L_{m+n} \right] + \mathcal{N}^{-1}_{(m,n),(m+n)} L_{-(m+n)}\rangle\langle L_{(m,n)}$$
$$+ \mathcal{N}^{-1}_{(m+n),(m+n)} \, L_{-(m+n)}\rangle\langle L_{(m+n)} \tag{3.38}$$

where $\mathcal{N}^{-1}_{i,i}$ in $I_1$ goes as $1/c$ while the $\mathcal{N}^{-1}$s in $I_2$ all go as $1/c^2$. Above we have computed the connected contribution to (D.1) at $1/c^2$ coming from

$$\widetilde{V}^{(6),Vir}_{T,comb} = \frac{\langle X_3 X_5 \,|I_1|\, \phi_1\, \phi_2 \,|I_1|\, Y_4 Y_6\rangle}{\langle\phi_1\phi_2\rangle\langle X_3 X_5\rangle\langle Y_4 Y_6\rangle} \sim \mathcal{O}(1/c^2) \tag{3.39}$$

The other contributions to the connected diagram at this order can be shown to be obtained by exchanging $\phi$s with $X$s and $\phi$s with $Y$s. We show this in appendix-D. Therefore the full contribution of the vacuum block to the connected 6pt comb channel (D.1) at $1/c^2$ is

$$V^{(6)\mathrm{Vir}}_{T,\mathrm{comb}} = \frac{\langle X_3 X_5 \,|\mathbb{I}|\, \phi_1\, \phi_2 \,|\mathbb{I}|\, Y_4 Y_6\rangle}{\langle\phi_1\phi_2\rangle\langle X_3 X_5\rangle\langle Y_4 Y_6\rangle}\bigg|^{\mathrm{conn}}_{c^{-2}} = \frac{\langle X_3 X_5 \,|I_1|\, \phi_1\, \phi_2 \,|I_1|\, Y_4 Y_6\rangle}{\langle\phi_1\phi_2\rangle\langle X_3 X_5\rangle\langle Y_4 Y_6\rangle} + \frac{\langle \phi_1\, \phi_2 \,|I_1|\, X_3 X_5 \,|I_1|\, Y_4 Y_6\rangle}{\langle\phi_1\phi_2\rangle\langle X_3 X_5\rangle\langle Y_4 Y_6\rangle} +$$
$$+ \frac{\langle X_3 X_5 \,|I_1|\, Y_4 Y_6 \,|I_1|\, \phi_1\, \phi_2\rangle}{\langle\phi_1\phi_2\rangle\langle X_3 X_5\rangle\langle Y_4 Y_6\rangle} \tag{3.40}$$

Therefore we have the full expression upto $\mathcal{O}(1/c^2)$ order as

$$\boxed{V^{(6)\mathrm{Vir}}_{T,\mathrm{comb}} = h_X h_Y \frac{\langle \mathcal{B}^{(1)}_{35} \mathcal{B}^{(1)}_{46}\phi_1\phi_2\rangle'}{\langle\phi_1\phi_2\rangle} + h_\phi h_Y \frac{\langle \mathcal{B}^{(1)}_{12} \mathcal{B}^{(1)}_{46} X_3 X_5\rangle'}{\langle X_3 X_5\rangle} + h_\phi h_X \frac{\langle \mathcal{B}^{(1)}_{35} \mathcal{B}^{(1)}_{12} Y_4 Y_6\rangle'}{\langle Y_4 Y_6\rangle} + \mathcal{O}(1/c^3)} \tag{3.41}$$

### 3.3.1 Simpler 8pt and 9pt functions in comb channel

We will next extend the method used for computing the above new result for the 6pt stress-tensor comb channel to that of 8pt and 9pt comb channels of a specific kind. We note these as the method used to compute the 6$pt$. case above readily allows for the computation of such functions too. It remains to be seen if they could be of any use.

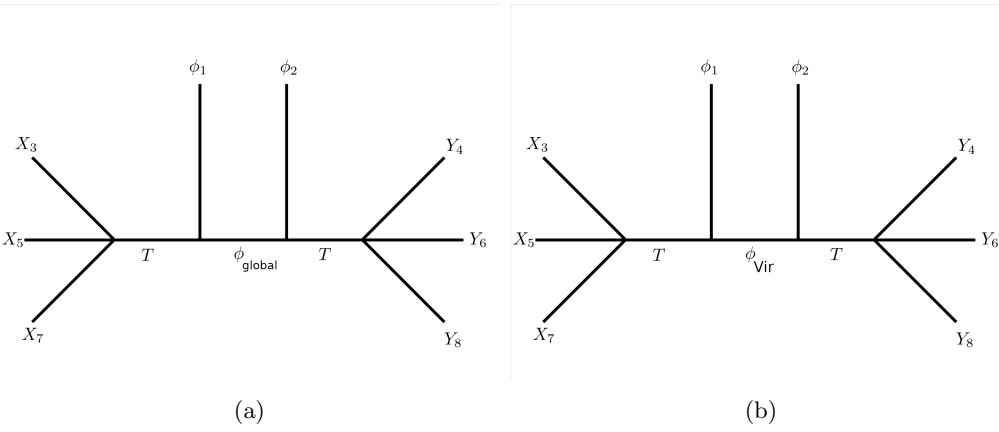

(a)                                          (b)

**Figure 2**: (a) Global part of the stress-tensor comb channel. (b)Virasoro block of the stress-tensor comb channel. The fusion of the propagators $X_{\{3,5,7\}}$ and $Y_{\{4,6,8\}}$ with the stress-tensor is entirely fixed by the stress-tensor Ward identity.

Simpler generalizations of the comb channel can be constructed by considering triplets of identical operators in place of identical pairs, Figure-2. The simplest of these is when there are 3 of each of the primary

operators $X$ and $Y$ and a pair of $\phi$s. Here too the global contribution is given by

$$\int_{0,-1} \frac{\langle X_{3,5,7}T_0\rangle\langle\widetilde{T}_0\phi_1|\phi_g|\phi_2\widetilde{T}_{-1}\rangle\langle T_{-1}Y_{4,6,8}\rangle}{\langle X_{3,5,7}\rangle\langle Y_{4,6,8}\rangle\langle\phi_{1,2}\rangle}, \tag{3.42}$$

Figure-2(a), with the internal $|\phi_g|$ projecting onto global states. The corresponding Virosoro contribution is obtained by allowing all states to propagate instead of the ones allowed by $|\phi_g|$

$$\int_{0,-1} \frac{\langle X_{3,5,7}T_0\rangle\langle\widetilde{T}_0\phi_{1,2}\widetilde{T}_{-1}\rangle\langle T_{-1}Y_{4,6,8}\rangle}{\langle X_{3,5,7}\rangle\langle Y_{4,6,8}\rangle\langle\phi_{1,2}\rangle}, \tag{3.43}$$

Figure-2(b). The obvious change here is the fusion of $X_{3,5,7}$ and $Y_{4,6,8}$ into the stress-tensor. These are easily determined as global conformal invariance fixes both the 2pt & 3pt functions of primaries.

One can also similarly consider a comb channel wherein there are 3 identical $\phi$ operators instead of 2. In this case the global contribution of to a similar comb channel can be written as

$$\int_{-2,-1} \frac{\langle X_{3,5,7}T_{-2}\rangle\langle\widetilde{T}_{-2}\phi_1|\phi_g|\phi_2|\phi_g|\phi_0\widetilde{T}_{-1}\rangle\langle T_{-1}Y_{4,6,8}\rangle}{\langle X_{3,5,7}\rangle\langle Y_{4,6,8}\rangle\langle\phi_{1,2,0}\rangle} \tag{3.44}$$

depicted as Figure-3(a).

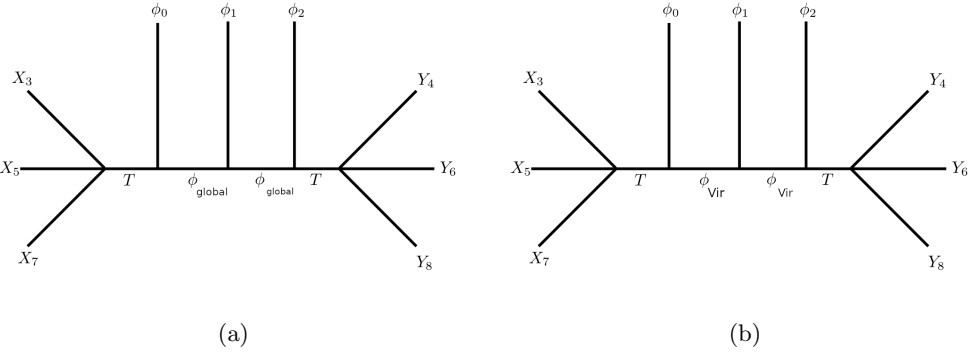

(a)                      (b)

**Figure 3**: (a) Global part of the stress-tensor comb channel. (b)Virasoro block of the stress-tensor comb channel. As the fusion of $T\phi \to \phi + \phi_{desc}$ the only states that can propagate between the $\phi_i$s in $\langle\widetilde{T}\phi_1\phi_2\phi_0\widetilde{T}\rangle$ are the Virasoro descendants of $\phi$.

The Virasoro contribution to such a comb channel is to remove all the insertions of $|\phi_g|$

$$\int_{-2,-1} \frac{\langle X_{3,5,7}T_{-2}\rangle\langle\widetilde{T}_{-2}\phi_1\phi_2\phi_0\widetilde{T}_{-1}\rangle\langle T_{-1}Y_{4,6,8}\rangle}{\langle X_{3,5,7}\rangle\langle Y_{4,6,8}\rangle\langle\phi_{1,2,0}\rangle} \tag{3.45}$$

we depict this as Figure-3(b).

$$\tag{3.46}$$

We will see in this subsection that both (3.43) & (3.45) can be computed using the expressions already evaluated thus far.

To compute (3.43) we simply need to generalize the results of the previous expression (3.25) as follows

$$\widetilde{V}^{(8),Vir}_{T,comb} = \frac{\langle X_{3,5,7}\,|T|\,\phi_{1,2}\,|T|\,Y_{4,6,8}\rangle}{\langle\phi_{1,2}\rangle\langle X_{3,5,7}\rangle\langle Y_{4,6,8}\rangle} = \left(\frac{3}{\pi c}\right)^2 \int_{3',4'} \frac{\langle\widetilde{T}_{3'}\widetilde{T}_{4'}\phi_{1,2}\rangle\langle T_{3'}X_{3,5,7}\rangle\langle T_{4'}Y_{4,6,8}\rangle}{\langle\phi_{1,2}\rangle\langle X_{3,5,7}\rangle\langle Y_{4,6,8}}$$

$$= h_X h_Y \frac{\left\langle \mathcal{B}_{357}^{(1)} \mathcal{B}_{468}^{(1)} \phi_1 \phi_2 \right\rangle'}{\langle \phi_1 \phi_2 \rangle} \tag{3.47}$$

wherein we introduce $\mathcal{B}_{123}^{(1)}$. In going from the first line on the *rhs* to the next we made use of the fact that $\widetilde{T} = \frac{c}{3}\bar{\partial}\epsilon$ and $\bar{\partial}T \equiv \partial^\mu T_{\mu\nu}$ and use the Ward identity for $\bar{\partial}_{3'}\langle T_{3'} X_{3,5,7} \rangle$ to write

$$\epsilon_{3'} \bar{\partial}_{3'} \langle T_{3'} X_{3,5,7} \rangle \equiv \mathcal{B}_{357}^{(1)} \langle X_{3,5,7} \rangle \tag{3.48}$$

$\mathcal{B}_{123}^{(1)}$ can be thus be determined from the 3pt. Ward identity as follows

$$\langle T_0 \phi_1 \phi_2 \phi_3 \rangle = \sum_{i-1}^{3} \left( \frac{h_\phi}{z_{0i}^2} + \frac{\partial_i}{z_{0i}} \right) \langle \phi_1 \phi_2 \phi_3 \rangle = \frac{h_\phi}{2} \left\langle (\mathcal{B}_{12}^{(1)} + \mathcal{B}_{23}^{(1)} + \mathcal{B}_{31}^{(1)}) T_0 \right\rangle \langle \phi_1 \phi_2 \phi_3 \rangle$$

$$=: h_\phi \langle \mathcal{B}_{123}^{(1)} T_0 \rangle \langle \phi_{1,2,3} \rangle$$

$$\implies \langle \epsilon_0 \phi_{1,2,3} \rangle = \frac{h_\phi}{2} \left\langle \left( \mathcal{B}_{12}^{(1)} + \mathcal{B}_{23}^{(1)} + \mathcal{B}_{13}^{(1)} \right) \epsilon_0 \right\rangle \langle \phi_{1,2,3} \rangle = \frac{h_\phi}{2} \langle \mathcal{B}_{123}^{(1)} \epsilon_0 \rangle \langle \phi_{1,2,3} \rangle \tag{3.49}$$

*i.e.*

$$\mathcal{B}_{123}^{(1)} = \frac{1}{2}(\mathcal{B}_{12}^{(1)} + \mathcal{B}_{23}^{(1)} + \mathcal{B}_{13}^{(1)}) \tag{3.50}$$

This is identical to the expression used earlier in the paper for $\mathcal{B}_{12}^{(1)}$ *i.e.*

$$\langle T_0 \phi_{1,2} \rangle = h_\phi \langle \mathcal{B}_{12}^{(1)} T_0 \rangle \langle \phi_{1,2} \rangle \tag{3.51}$$

Just as the sub-leading 4pt vacuum block of 2 pairs of identical operators is given by

$$\frac{\langle \phi_{1,2} \psi_{3,4} \rangle}{\langle \phi_{1,2} \rangle \langle \psi_{3,4} \rangle} = 1 + h_\phi h_\psi \langle \mathcal{B}_{12}^{(1)} \mathcal{B}_{34}^{(1)} \rangle \tag{3.52}$$

The sub-leading contributions to $\langle \phi_{1,2,3} \psi_{4,6} \rangle$ and $\langle \phi_{1,2,3} \psi_{4,6,8} \rangle$ are likewise given by

$$\frac{\langle \phi_{1,2,3} \psi_{4,6} \rangle}{\langle \phi_{1,2,3} \rangle \langle \psi_{4,6} \rangle} = 1 + h_\phi h_\psi \langle \mathcal{B}_{123}^{(1)} \mathcal{B}_{46}^{(1)} \rangle, \qquad \frac{\langle \phi_{1,2,3} \psi_{4,6,8} \rangle}{\langle \phi_{1,2,3} \rangle \langle \psi_{4,6,8} \rangle} = 1 + h_\phi h_\psi \langle \mathcal{B}_{123}^{(1)} \mathcal{B}_{468}^{(1)} \rangle \tag{3.53}$$

which are obtained by repeating the manipulations outlined above. Therefore the Virasoro contribution to the comb channel (3.43) can be obtained by knowing $\langle \mathcal{B}_{35}^{(1)} \mathcal{B}_{46}^{(1)} \phi_{1,2} \rangle'$ in (3.26) where the prime indicates that we ignore the contact term (first term) in (3.26).

In order to compute (3.45) we would have to find the expression for $\langle \epsilon_{-1} \epsilon_{-2} \phi_{0,1,2} \rangle$. We make use of the Ward identity for $\langle T_{-1} T_{-2} \phi_{0,1,2} \rangle$ to find this. We note that the Ward identity $\langle T_{-1} T_{-2} \phi_{0,1,2} \rangle$ can be written as

$$\frac{\langle T_{-1} T_{-2} \phi_{0,1,2} \rangle'}{\langle \phi_{0,1,2} \rangle} = \frac{1}{2} \left\{ h_\phi(h_\phi - 1) \langle \mathcal{B}_{01}^{(1)} T_{-1} \rangle \langle \mathcal{B}_{01}^{(1)} T_{-2} \rangle + h \langle \mathcal{B}_{01}^{(1)} T_{-1} \rangle \left( \langle \mathcal{B}_{-10}^{(1)} T_{-2} \rangle + \langle \mathcal{B}_{-11}^{(1)} T_{-2} \rangle \right) \right.$$

$$\left. + \frac{h^2}{2} \langle \mathcal{B}_{-10}^{(1)} T_{-1} \rangle \langle \left( \mathcal{B}_{12}^{(1)} + \mathcal{B}_{02}^{(1)} - \mathcal{B}_{01}^{(1)} \right) T_{-2} \rangle + \text{cyclic}_{\{0,1,2\}} \right\}$$

$$\text{thus} \implies \frac{\langle \epsilon_{-1} \epsilon_{-2} \phi_{0,1,2} \rangle'}{\langle \phi_{0,1,2} \rangle} = \frac{1}{2} \left\{ h(h-1) \langle \mathcal{B}_{01}^{(1)} \epsilon_{-1} \rangle \langle \mathcal{B}_{01}^{(1)} \epsilon_{-2} \rangle + h \langle \mathcal{B}_{01}^{(1)} \epsilon_{-1} \rangle \left( \langle \mathcal{B}_{-10}^{(1)} \epsilon_{-2} \rangle + \langle \mathcal{B}_{-11}^{(1)} \epsilon_{-2} \rangle \right) \right.$$

$$\left. + \frac{h^2}{2} \langle \mathcal{B}_{10}^{(1)} \epsilon_{-1} \rangle \langle \left( \mathcal{B}_{12}^{(1)} + \mathcal{B}_{02}^{(1)} - \mathcal{B}_{01}^{(1)} \right) \epsilon_{-2} \rangle + \text{cyclic}_{\{0,1,2\}} \right\} \tag{3.54}$$

where $\text{cyclic}_{\{0,1,2\}}$ implies we add exchanges of $\{0 \to 1 \to 2 \to 0\}$ and $\{0 \leftarrow 1 \leftarrow 2 \leftarrow 0\}$. Therefore writing (3.45) as

$$\widetilde{V}_{T,comb}^{(9),Vir} = \int_{-2,-1} \frac{\langle X_{3,5,7} T_{-2} \rangle \langle \widetilde{T}_{-2} \phi_1 \phi_2 \phi_0 \widetilde{T}_{-1} \rangle \langle T_{-1} Y_{4,6,8} \rangle}{\langle X_{3,5,7} \rangle \langle Y_{4,6,8} \rangle \langle \phi_{1,2,0} \rangle} = \frac{c^2}{9} \int_{-2,-1} \frac{\langle \epsilon_{-1} \epsilon_{-2} \phi_{0,1,2} \rangle \bar{\partial}_{-2} \langle T_{-2} X_{3,5,7} \rangle \bar{\partial}_{-1} \langle T_{-1} Y_{4,6,8} \rangle}{\langle X_{3,5,7} \rangle \langle Y_{4,6,8} \rangle \langle \phi_{1,2,0} \rangle}$$

$$= h_X h_Y \frac{\langle \mathcal{B}_{357}^{(1)} \mathcal{B}_{468}^{(1)} \phi_{0,1,2} \rangle'}{\langle \phi_{0,1,2} \rangle} \tag{3.55}$$

where the last expression is obtained by replacing $\epsilon_{-1}$ & $\epsilon_{-2}$ in (3.54) with $\mathcal{B}_{357}^{(1)}$ & $\mathcal{B}_{468}^{(1)}$ respectively, with some manipulations we can concisely write it as

$$h_X h_Y \frac{\langle \mathcal{B}_{357}^{(1)} \mathcal{B}_{468}^{(1)} \phi_{0,1,2} \rangle'}{\langle \phi_{0,1,2} \rangle} =$$
$$= \frac{h_X h_Y}{8} \left( \frac{\langle \mathcal{B}_{357}^{(1)} \mathcal{B}_{468}^{(1)} \phi_0 \phi_1 \rangle'}{\langle \phi_0 \phi_1 \rangle} + \frac{h_\phi^2}{2} \langle \mathcal{B}_{357}^{(1)} \mathcal{B}_{01}^{(1)} \rangle \left( \langle \mathcal{B}_{468}^{(1)} \mathcal{B}_{02}^{(1)} \rangle + \langle \mathcal{B}_{468}^{(1)} \mathcal{B}_{12}^{(1)} \rangle - \langle \mathcal{B}_{468}^{(1)} \mathcal{B}_{01}^{(1)} \rangle \right) + \text{cyclic}_{(0,1,2)} \right) \tag{3.56}$$

It is important to note that although cumbersome, the task of finding the above higher point functions has become entirely algebraic once the expression for $\langle \mathcal{B}_{ij}^{(1)} \mathcal{B}_{kl}^{(1)} \phi_1 \phi_2 \rangle$ or $\langle \epsilon_i \epsilon_k \phi_1 \phi_2 \rangle$ has ben determined. We summarize these functions in terms of cross ratios in the Appendix E.

As in the previous section for the $6pt.$ comb channel this is not the complete $\mathcal{O}(1/c^2)$ contribution to the $8pt.$ and $9pt.$ functions. For the case of the $9pt.$ function consider the insertion of complete projector onto the vacuum block $\mathbb{I}$ (3.38)

$$V_{T,comb}^{(9),Vir} = \frac{\langle X_{3,5,7} | \mathbb{I} | \phi_{0,1,2} | \mathbb{I} | Y_{4,6,8} \rangle}{\langle X_{3,5,7} \rangle \langle \phi_{0,1,2} \rangle \langle Y_{4,6,8} \rangle} \bigg|_{\mathcal{O}(c^{-2})} = \widetilde{V}_{T,comb}^{(9),Vir} + \widetilde{V}_{T,comb}^{(9),Vir} \bigg|_{(\phi_{0,1,2} \leftrightarrow X_{3,5,7})} + \widetilde{V}_{T,comb}^{(9),Vir} \bigg|_{(\phi_{0,1,2} \leftrightarrow Y_{4,6,8})} \tag{3.57}$$

where we use the expression (3.55) for $\widetilde{V}_{T,comb}^{(9),Vir}$ and symmetrize with respect to the all the operator triplets to account for other contribution from $I_2$ in $\mathbb{I}$. For the $8pt.$ case analysis similar to the $6pt.$ case leads to

$$V_{T,comb}^{(8),Vir} = \frac{\langle X_{3,5,7} | \mathbb{I} | \phi_{1,2} | \mathbb{I} | Y_{4,6,8} \rangle}{\langle X_{3,5,7} \rangle \langle \phi_{0,1,2} \rangle \langle Y_{4,6,8} \rangle} \bigg|_{\mathcal{O}(c^{-2})} = \widetilde{V}_{T,comb}^{(8),Vir} + h_\phi h_X \frac{\langle \mathcal{B}_{1,2}^{(1)} \mathcal{B}_{3,5,7}^{(1)} Y_{4,6,8} \rangle'}{\langle Y_{4,6,8} \rangle} + h_\phi h_Y \frac{\langle \mathcal{B}_{1,2}^{(1)} \mathcal{B}_{4,6,8}^{(1)} X_{3,5,7} \rangle'}{\langle X_{3,5,7} \rangle} \tag{3.58}$$

where we use (3.47) for $\widetilde{V}_{T,comb}^{(8),Vir}$. The analysis of the contribution from $I_2$ in $\mathbb{I}$ to the $8pt.$ & $9pt.$ case considered here is similar to the $6pt.$ case and is covered in the Appendix-D.

The higher pt. comb channels considered here (Fig-2 & Fig-3) consist of $4pt.$ vertices formed by the fusion of identical triplets with the stress-tensor in contrast with the $3pt.$ vertices one generally considered for channel decomposition. However such $4pt$ vertices can in turn be expanded uniquely into a diagram consisting of 2 $3pt$ vertices as shown below.

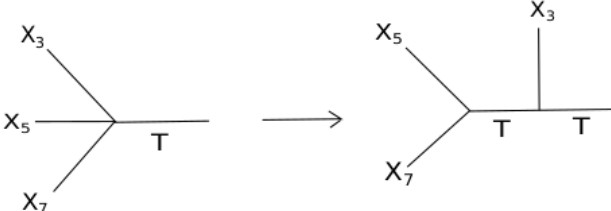

**Figure 4**: $4pt$ vertex formed from fusion of any $3pt$ funciton with stress-tensor can be broken up into the right-hand diagram consisting of two $3pt$ vertices uniquely.

Thus using the above substitution into the the higher $pt$ comb channels considered in Fig-2 & Fig-3, we see that these can be considered as comb channels with a particular choice of intermediate operators in

the internal legs[20]. It must also be noted that having replaced the 4*pt.* vertices in Fig-2 & Fig-3 as above, demanding that the operators on either ends fuse to give the stress-tensor[21] uniquely fixes the operators propagating the internal legs.

# 4 OTOCs

In this section we turn to computing various *out-time-ordered* correlators as a diagnostic of chaotic behaviour. We first compute the maximally braided OTOC for the Virasoro contribution of the 6pt comb channel. It was already shown in [36] that the global analogue of this correlator grows exponentially, we find that the full Virasoro correlator still exhibits the same behaviour. We next consider an *out-time-ordering* in which only the $X_{\{3,5\}}$ and $Y_{\{4,6\}}$ operator pairs are *out-time-ordered* while each of them are time ordered *w.r.t* the pair $\phi_{\{1,2\}}$. This particular *out-time-ordering* is special as the global stress tensor 6pt comb channel is seen not to grow exponentially. We also then consider the OTOC for $\langle \phi_{1,2,3} \psi_{3,4,5} \rangle$ and comment on similar generalizations obtained previously.

## 4.1 OTOC for $\langle \mathcal{B}\mathcal{B}\phi\phi \rangle_{c^{-2}}$

We next turn to finding the OTOC for the expression for (3.36) or equivalently the $\mathcal{O}(c^{-2})$ terms in (3.26). The full 6*pt.* stress-tensor comb channel (3.41) consists of 2 other terms obtained by symmetrization, however the contribution of the first term would be most important of the maximally braided *out of time ordering* considered here. In order to measure the Lyapunov index in a thermal background one needs to first map the line coordinate to a circle *via* the exponential map

$$z_i = e^{\frac{2\pi}{\beta}(t_i + \sigma_i - i\tau_i)} \tag{4.1}$$

where we have also retained the Euclidean time $\tau$. We set the Lotrentzian times $t_i=0$ and choose a specific Euclidean time ordering for the various points in (3.26) as

$$\tau_4 < \tau_2 < \tau_6 < \tau_5 < \tau_1 < \tau_3 \tag{4.2}$$

for the out of time ordered case, and

$$\tau_4 < \tau_6 < \tau_2 < \tau_1 < \tau_5 < \tau_3 \tag{4.3}$$

for the time ordered case. We will set the spatial points to

$$\sigma_4 = \sigma_6 = \sigma_Y \quad > \quad \sigma_1 = \sigma_2 = \sigma_\phi \quad > \quad \sigma_3 = \sigma_5 = \sigma_X. \tag{4.4}$$

Having evaluated the Euclidean answer we next turn on the Loretnzian times with the following ordering

$$t_4 = t_6 = t_Y \quad < \quad t_1 = t_2 = t_\phi \quad < \quad t_3 = t_5 = t_X \tag{4.5}$$

Here we choose to work with the following invariant cross ratios

$$z = \frac{z_{12}z_{34}}{z_{13}z_{24}}, \qquad y = \frac{z_{12}z_{56}}{z_{15}z_{26}}, \qquad u = \frac{z_{12}z_{54}}{z_{15}z_{24}}. \tag{4.6}$$

the Regge limit for whom would be $z \to 0, y \to 0, u \to 0$.[22] The final expression for the various terms in (3.26) would consist of logarithms and di-logarithms(Li$_2$) of functions of the above cross ratios. These (di-)logarithms have branch cuts in their arguments[23]. As the cross ratios approach the Regge limit they trace

---

[20]*Note,* that the internal stress-tensor in the right-hand diagram in Fig-4 denotes the the exchange of all possible states descended from the vacuum as the 3*pt.* function- like the 2*pt.* function, on the left-hand side is completely fixed by global conformal symmetries.

[21]This can be taken to be the general definition of the stress-tensor comb channel block.

[22]It might be useful to switch to $Z = \frac{z_{31}z_{25}}{z_{32}z_{15}} = \frac{1-u}{1-z}$, $U = \frac{z_{31}z_{24}}{z_{32}z_{14}} = \frac{1}{1-z}$, $V = \frac{z_{31}z_{26}}{z_{32}z_{16}} = \frac{1-u}{(1-y)(1-z)}$ which tend to 1 in the Regge limit. The behaviour of these cross ratios for the out of time ordering of (4.2) is plotted in [36].

[23]$\log x$ has a branch cut from $x \in (-\infty, 0]$ with a discontinuity of $2\pi i$, while Li$_2 x$ has a branch cut from $x \in [1, \infty)$ and it picks up a value of $2\pi i \log x$

a contour in their complex plane and depending on the time ordering and their functional dependence inside the logarithm and di-logarithms they may or may not cross these branch cuts. The contours traced by the cross ratios for TO correlators is such that the they do not receive any contribution from these branch cuts. For OTO correlators however the contours traced do cross certain branch cuts and it is these contributions which give the exponential behaviour.

### 4.1.1 OTO of $\langle \mathcal{BB} \rangle \langle \mathcal{BB} \rangle$

We fist look that OTO behaviour of the second term in (3.26) as this seems like a square of the 4pt stress-tensor block. In the Regge limit of (4.6) the relevant cross ratios involved in this term also tend to zero *i.e.*

$$\frac{z_{12}z_{35}}{z_{13}z_{25}} = \frac{u-z}{u-1} \to 0, \qquad \frac{z_{12}z_{46}}{z_{14}z_{26}} = \frac{u-y}{u-1} \to 0 \tag{4.7}$$

The branch cuts of Logarithms involved are crossed by the above cross ratios only for out of time ordering (4.2) and not for time ordered arrangement of (4.3). The exponentially growing pieces in the second term in (3.26) are as

$$h_\phi(h_\phi - 1) \left\{ \frac{\beta^4}{4c^2\pi^2\tau_{46}\tau_{12}^2\tau_{35}} \sinh^2\left(\frac{\pi(t_{\phi Y} - \sigma_{Y\phi})}{\beta}\right) \sinh^2\left(\frac{\pi(t_{X\phi} - \sigma_{\phi X})}{\beta}\right) \right.$$
$$\left. + \frac{i\beta^2}{2\pi c^2\tau_{12}\tau_{35}} \sinh^2\left(\frac{\pi(t_{X\phi} - \sigma_{\phi X})}{\beta}\right) + \frac{i\beta^2}{2\pi c^2\tau_{46}\tau_{12}} \sinh^2\left(\frac{\pi(t_{\phi Y} - \sigma_{Y\phi})}{\beta}\right) \ldots \right\} \tag{4.8}$$

where the only the first term is relevant for the growth in the largest time interval $t_{XY}$. Above we have only retained the leading terms in $\tau_{ij}$[24] in $1/c^2$ as the Euclidean time differences $\tau_{ij} \to 0$. We clearly see the behaviour obtained in [36] where the Lyapunov index for large $t_{XY} \gg \frac{\beta}{2\pi}$ is

$$\lambda_L = \frac{2\pi}{\beta} \tag{4.9}$$

with the exponential growth lasting for $2t^*$ where $t^* = \frac{\beta}{2\pi} \log(c)$ is the scrambling time. The first 2 terms above showcase the characteristic growth of the 4pt OTOC for respective large intermediate times $t_{X\phi}$ & $t_{\phi Y}$ with the same $\lambda_L$ but lasting for $t^*$. It is worth noting that to deduce the exponential behaviour in all the time intervals above we did not have to solve any non-trivial differential equations. However for operator dimension $h_\phi \ll c$, the linear in $h_\phi$ terms in the above expression do combine with the exponential growth coming from OTO behaviour of the last term in (3.26) which we next turn to.

### 4.1.2 OTO of $\langle \mathcal{K}^{(2)} \rangle$

This term is linear in $h_\phi$ and has terms which are proportional to $\text{Li}_2, \log$ & $\log^2$ of various cross ratios. The leading contribution in the Regge limit for OTO placement of operators (4.2) is of the form

$$\langle \mathcal{K}^{(2)} \rangle_{\text{oto}} \approx \frac{11\beta^4}{4\tau_{46}\tau_{12}^2\tau_{35}} \sinh^2\left(\frac{\pi(t_{\phi Y} - \sigma_{Y\phi})}{\beta}\right) \sinh^2\left(\frac{\pi(t_{X\phi} - \sigma_{\phi X})}{\beta}\right) + \mathcal{O}(\tau_{ij}^{-3}) \tag{4.10}$$

which is the leading order behaviour for $t_{XY} \gg \frac{\beta}{2\pi}$[25]. Moreover the exponential behaviours for time interval $t_{XY}$ of terms linear in operator dimension $h$ persist upon adding the relevant terms from (4.8).

$$\frac{\left\langle \mathcal{B}_{35}^{(1)} \mathcal{B}_{46}^{(1)} \phi_1 \phi_2 \right\rangle}{\langle \phi_1 \phi_2 \rangle} \approx \frac{1085\beta^4 h_\phi}{c^2\pi^2\tau_{46}\tau_{12}^2\tau_{35}} \sinh^2\left(\frac{\pi(t_{\phi Y} - \sigma_{Y\phi})}{\beta}\right) \sinh^2\left(\frac{\pi(t_{X\phi} - \sigma_{\phi X})}{\beta}\right) + \mathcal{O}(\tau_{ij}^{-3}) \tag{4.11}$$

---

[24]$\tau_{ij} = \tau_i - \tau_j$

[25]Here we have suppressed terms of order $\mathcal{O}(\tau_{12}^{-3})$

Here we have retained the only the most singular terms as $\tau_{ij} \to 0$[26]. We thus see a growth in the largest time interval $t_{XY}$ governed by a Lyapunov index

$$\lambda_L = \frac{2\pi}{\beta} \tag{4.12}$$

with a growth lasting for $2t^* = \frac{\beta}{\pi} \log c$. This bodes well with the expectation that $2n$-pt function without of time ordering such that one requires $n-1$ turns in the complex time plane to faithfully describe it, grows exponentially for [19]

$$(n-1)t^* \tag{4.13}$$

However this was proved to hold [19] for a 1d theory of reparametrizations governed by the Schwarzian action. This exact behaviour was also deduced in the case of 6pt star channel vacuum block in CFT$_2$ in [36].

The full $6pt$ Virasoro stress-tensor comb channel at $\mathcal{O}(1/c^2)$ (3.41) is basically symmetric with respect to all the 3 pairs of operators. However, the maximally braided *out of time ordering* considered here (4.2)(4.5) has the $X$ and $Y$ pairs *time ordered w.r.t.* each other. Therefore the rest of the terms would give a sub-leading behaviour as compared to (4.11) in terms of the time intervals considered in (4.5).

## 4.2 4pt. OTO of $X_{\{3,5\}}$ and $Y_{\{4,6\}}$ in 6pt comb channel

In this subsection we consider the *out-of-time-ordering* of only the $X_{\{3,5\}}$ and the $Y_{\{4,6\}}$ pairs with each other while each of them being time ordered *w.r.t.* the $\phi_{\{1,2\}}$ pair. We work with the following cross ratios

$$Z = \frac{z_{31} z_{25}}{z_{32} z_{15}} = \frac{1-u}{1-z}, \ U = \frac{z_{31} z_{24}}{z_{32} z_{14}} = \frac{1}{1-z}, \ V = \frac{z_{31} z_{26}}{z_{32} z_{16}} = \frac{1-u}{(1-y)(1-z)} \tag{4.14}$$

in this subsection. We choose the following Euclidean times for all Lorentzian time stamps set to zero initially

$$\tau_1 < \tau_2 < \tau_4 < \tau_3 < \tau_6 < \tau_5 \tag{4.15}$$

with the spatial coordinate $\sigma_i$s ordered as

$$\sigma_\phi = \sigma_1 = \sigma_2 = 2\sigma, \ \sigma_Y = \sigma_4 = \sigma_6 = \sigma, \ \sigma_X = \sigma_3 = \sigma_5 = 0, \quad \text{with } \sigma > 0 \tag{4.16}$$

Having evaluated the Euclidean correlator we increase the Lorentzian times as

$$t_\phi = t_1 = t_2 = 0, \ t_Y = t_4 = t_6 = t, \ t_X = t_3 = t_5 = 2t, \quad \text{where } t > 0 \tag{4.17}$$

### 4.2.1 Global comb channel

We note the global stress-tensor comb channel's answer from [36] below.

$$\mathcal{I}(Z,U,V) = \frac{U}{2} \left( \frac{U+V}{U} - \frac{2V}{U-V} \log \frac{U}{V} \right) \left( \frac{1+Z}{Z} + \frac{2}{1+Z} \log Z \right) + h_\phi \left( 2 - \frac{U+V}{U-V} \log \frac{U}{V} \right) \left( 2 + \frac{1+Z}{1-Z} \log Z \right)$$

$$+ \frac{1}{2h_\phi + 1} \left\{ [F_1(U) + F_1(V)] - \frac{4UV}{(U-V)(1-Z)} [F_2(U) - F_2(V)] + \frac{(U+V)(1+Z)}{2(U-V)(1-Z)} [F_3(U) - F_3(V)] \right\} \tag{4.18}$$

where $F_i$s are given in terms of Hypergeometric functions as

$$\begin{aligned} F_1(x) &= x^2 {}_2F_1(1,1;2h_\phi + 2; x), \\ F_2(x) &= x {}_3F_2(1,1,1;2,2h_\phi + 2; x), \\ F_3(x) &= x^2 {}_3F_2(1,1,2;3,2h_\phi + 2; x). \end{aligned} \tag{4.19}$$

---

[26] The sub-leading terms $1/\tau_{ij}$ in the limit $\tau_{ij} \to 0$ also exhibit the exact same behaviour.

We also note that this answer unlike the full Virasoro answer is not automatically symmetric *w.r.t.* the 3 pairs of operators. To be precise the answer above corresponds to the Fig-1a. One can compare the Virasoro answer (3.41) or (3.36) and see that the global answer above is not easily discernable in it. This could be due to the apparent difference in the method of computation as the above answer was obtained by inserting the projector onto the global descendants of $\phi$ between the 2 $\phi$s[27] while the Virasoro contribution simply makes use of the $2pt$ function of $\phi$s and their Ward identities.

We contrast the behaviour of above *out-of-time-ordering* in the Virasoro comb channel with that of the global comb channel answer for the same configuration. It can be seen that for the OTO generated by (4.15)(4.17)(4.16) only the first line in (4.18) is relevant as the hypergeometrics in line 2 of (4.18) do not contribute any divergent terms in the Regge limit of $\{Z, U, V\} \to 1$. Also we observe that no branch cuts of the logarithms in the first line of the expression (4.18) are crossed for the above OTO. Thus the global comb channel does not show any exponentially growing behaviour for the above *out-of-time-ordering*.

This can be thought of as a consequence of only global states associated with the operator $\phi$ propagating in the internal leg of the comb channel. In a holographic setting where the operators in the CFT are dual to bulk fields of appropriate spin, the global states can be thought of generated by Killing isometries of the ambient bulk black hole in $AdS$ as these are dual to the global conformal symmetries of the boundary CFT. Therefore it is not surprising from such a holographic perspective to expect no exponential growth for large time separations between the $X$ and $Y$ pairs for the above out of time ordering as there are no gravitons being propagated between the pairs in the internal leg. These gravitons as seen in the bulk picture are captured by Virasoro states associated with the operator $\phi$ in the internal leg. Therefore as such a contribution is captured in the Virasoro comb channel we may expect to see a different behaviour for the above out of time ordering.

### 4.2.2 Virasoro Comb channel

We next consider the Virasoro comb channel answer obtained in (3.41). We evaluate the contribution in parts as we can associate diagrams with each of them.

$$V_{T,\text{comb}}^{(6)\text{Vir}} = h_X h_Y \frac{\langle \mathcal{B}_{35}^{(1)} \mathcal{B}_{46}^{(1)} \phi_1 \phi_2 \rangle'}{\langle \phi_1 \phi_2 \rangle} + h_\phi h_Y \frac{\langle \mathcal{B}_{12}^{(1)} \mathcal{B}_{46}^{(1)} X_3 X_5 \rangle'}{\langle X_3 X_5 \rangle} + h_\phi h_X \frac{\langle \mathcal{B}_{35}^{(1)} \mathcal{B}_{12}^{(1)} Y_4 Y_6 \rangle'}{\langle Y_4 Y_6 \rangle} + \mathcal{O}(1/c^3) \qquad (4.20)$$

The first term for example is the evaluation of (3.39) where $I_1$ depicts that level-1 states associated with the vacuum are being exchanged between $X_{3,5}$ & $\phi_{1,2}$ and $Y_{4,6}$ & $\phi_{1,2}$. As explained earlier Fig-1b the internal line for propagation of states generated by $\phi$ takes into account its Virasoro descendants.

The expression for $\langle \mathcal{B}_{3,5}^{(1)} \mathcal{B}_{4,6}^{(1)} \phi_1 \phi_2 \rangle$ is given by (3.26) where the first term is to be ignored as it does not contribute to the comb channel. The second term in (3.26) consists of a product of 4pt. vacuum blocks between $X_{3,5}$ & $\phi_{1,2}$ and $Y_{4,6}$ & $\phi_{1,2}$. As the above out of time ordering effects only the time ordering between $X_{3,5}$ & $Y_{4,6}$, the second term in (3.26) does not contribute any exponential behaviour. Therefore the only relevant term in Virasoro comb channel for the above out of time ordering is the last term in (3.26) *i.e.* $\mathcal{K}^{(2)}$.

As the expression for $\mathcal{K}^{(2)}$ is quite huge and it serves little purpose to spell it out in its entirety in terms of cross-ratios explicitly. However we note that it can be grouped into terms of 3 kinds, $\mathcal{K}^{(2)} = V_{\text{Li}} + V_{\log^2} + V_{\log}$ where:

    1  Terms linear in $Li_2$ - $V_{\text{Li}}$

    2  Terms quadratic in logarithms - $V_{\log^2}$

---

[27]*c.f.* eq-5.1 of [36].

## 3 Terms linear in logarithms - $V_{\log}$

The arguments of the $Li_2$s and logarithms are relevant in terms of their behaviour in the complex time plane while their coefficients are relevant as they would multiply the discontinuity across the branch cuts. For the logarithms the relevant terms which cross the branch cuts for the above out of time ordering are

$$\log\left(\frac{1-V}{Z-V}\right) \quad \& \quad \log\left(Z\,\frac{1-V}{Z-V}\right) \tag{4.21}$$

Terms proportional to the above logarithms contribute branch cut discontinuities in the terms $V_{\log^2}$ and $V_{\log}$. However in both $V_{\log^2}$ and $V_{\log}$ these discontinuities multiply functions which are finite in the Regge limit $\{Z,U,V\} \to 1$. Thus these terms do not contribute any exponential growth.

Similar terms relevant for $V_{\mathrm{Li}}$ are

$$\mathrm{Li}_2\left(\frac{1-Z}{1-V}\right) \quad \& \quad \mathrm{Li}_2\left(\frac{1-Z}{1-V}\frac{V}{Z}\right) \tag{4.22}$$

The net discontinuity across the branch cut contributes a finite term in the Regge limit. Thus even this term does not grow exponentially for the above out of time ordering. The OTOC is of the form

$$
\begin{aligned}
h_X h_Y & \frac{\langle \mathcal{B}_{35}^{(1)} \mathcal{B}_{46}^{(1)} \phi_1 \phi_2 \rangle'_{OTOC}}{\langle \phi_1 \phi_2 \rangle} \sim \\
& = \frac{\pi i 288 h_\phi h_X h_Y}{c^2} \left[ \frac{5h\left(U\left(-2VZ + Z^2 + 1\right) + VZ^2 + V - 2Z\right)}{8(Z-1)^2(U-V)} \log\left(\frac{Z}{V}\right) + \right. \\
& \left. + \frac{U\left(5V^2(2Z-1) + V\left(8Z^2 - 18Z + 5\right) - 3(Z-1)Z\right) + Z\left(-2V^2(Z+4) + V(2Z+13) - 5Z\right)}{8(V-1)(Z-1)Z(U-V)} \right]
\end{aligned}
\tag{4.23}
$$

which only tends to constant in the Regge limit. Therefore although various branch cuts are crossed by the terms in $\mathcal{K}^{(2)}$ their discontinuities multiply functions with finite behaviours in the Regge limit. Therefore, this particular contribution to the Virasoro comb channel does not show an exponentially growing behaviour for the large time OTO between the $X_{3,5}$ and $Y_{4,6}$ pairs.

We next consider the second term

$$\frac{\langle \mathcal{B}_{12}^{(1)} \mathcal{B}_{46}^{(1)} X_3 X_5 \rangle'}{\langle X_3 X_5 \rangle} = h_X(h_X - 1)\langle \mathcal{B}_{12}^{(1)} \mathcal{B}_{35}^{(1)} \rangle \langle \mathcal{B}_{35}^{(1)} \mathcal{B}_{46}^{(1)} \rangle + h_X\left(\frac{c}{12}\right)^2 \mathcal{K}_{35}^{(2)} \tag{4.24}$$

where we used (3.26) and $\mathcal{K}_{35}^{(2)}$ is $\mathcal{K}^{(2)}$ computed in the previous section with $\phi_{1,2} \leftrightarrow X_{3,5}$. The factor $\langle \mathcal{B}_{35}^{(1)} \mathcal{B}_{46}^{(1)} \rangle$ is the same function as the $\mathcal{O}(1/c)$ $4pt$ vacuum block contribution between $X_{35}$ and $Y_{46}$ which does grow exponentially when $X_{3,5}$ and $Y_{4,6}$ are out of time ordered with respect to each other. But this is multiplied with $\langle \mathcal{B}_{12}^{(1)} \mathcal{B}_{35}^{(1)} \rangle$ which vanishes for large time separations between $X_{3,5}$ and $\phi_{1,2}$. One can check that this term therefore does not grow exponentially for large time separations considered here. The same factor appears in the third term too

$$\frac{\langle \mathcal{B}_{12}^{(1)} \mathcal{B}_{35}^{(1)} Y_4 Y_6 \rangle'}{\langle Y_4 Y_6 \rangle} = h_Y(h_Y - 1)\langle \mathcal{B}_{12}^{(1)} \mathcal{B}_{46}^{(1)} \rangle \langle \mathcal{B}_{35}^{(1)} \mathcal{B}_{46}^{(1)} \rangle + h_Y\left(\frac{c}{12}\right)^2 \mathcal{K}_{46}^{(2)} \tag{4.25}$$

where $\mathcal{K}_{46}^{(2)}$ is similarly obtained by replacing $\phi_{1,2} \leftrightarrow Y_{4,6}$ in $\mathcal{K}^{(2)}$. Here too the first term in the above $rhs$ does not contribute an exponentially growing term.

It further turns out that the contributions from $\mathcal{K}^{(2)}_{35}$ and $\mathcal{K}^{(2)}_{46}$ above have behaviours similar to that of $\mathcal{K}^{(2)}$ for the out of time ordering (4.15)(4.16)(4.17) and there appears to be no exponentially growing terms.

This is in stark contrast with the behaviour between the global and Virasoro contributions to the 6pt. vacuum star channel considered in [36]. There the global star channel showed no exponential growth but the Virasoro contribution explicitly did.

This is a very interesting result if one were to consider the case of correlators with a pair of heavy operators in a thermal back ground. The simplest case is that of $\langle \Phi_1 \Phi_2 \psi_3 \psi_4 \rangle$ where the $\Phi$s have dimension $H \sim c$. The leading order term in the $1/c$ expansion is obtained by writing the power law 2pt function $\langle \psi_3 \psi_4 \rangle$ in the $w$ frame and performing a conformal transformation $w = z^\alpha$ where $\alpha = \sqrt{1 - \frac{24H}{c}}$. $z$ can be mapped to a unit circle $z = e^x$ or to a thermal circle $z = \exp\left[\frac{2\pi}{\beta} x\right]$. The leading order in the later case becomes [51]

$$\frac{\langle \Phi_1 \Phi_2 \psi_3 \psi_4 \rangle}{\langle \Phi_1 \Phi_2 \rangle} = \frac{1}{\sinh\left[\frac{\pi}{\widehat{\beta}} x_{34}\right]} + \mathcal{O}(1/c), \quad \text{where } \widehat{\beta} = \frac{\beta}{\sqrt{1 - \frac{24H}{c}}} \tag{4.26}$$

Here we placed $x_1$ at $\infty$ and $x_2$ at $-\infty$. As the presence of heavy operators modifies the temperature it is reasonable to expect that the 4pt OTO of light operators $X_{3,5}$ and $Y_{4,6}$ in such a background would grow like [51]

$$\frac{\langle \Phi_{1,2} X_{3,5}, Y_{4,6} \rangle}{\langle \Phi_{1,2} \rangle \langle X_{3,5} \rangle \langle Y_{4,6} \rangle}\bigg|_\beta \sim \exp\left[\frac{2\pi}{\widehat{\beta}}((t_X - t_Y) - (\sigma_X - \sigma_Y))\right] \tag{4.27}$$

If we then take the limit that the heavy $\Phi$s become light *i.e.* expand in $H/c$; we see that $\widehat{\beta} \to \beta$. Therefore the 4pt OTO operators $X_{3,5}$ and $Y_{4,6}$ in the 6pt function $\langle \phi_{1,2} X_{3,5}, Y_{4,6} \rangle$ (here $\Phi \to \phi$ to indicate they are light) grows with a Lyapunov index $\frac{2\pi}{\beta}$[28].

The 6pt vacuum star channel for light operators $\langle \phi_{1,2} X_{3,5}, Y_{4,6} \rangle^{\text{vac}}_{\text{star}}$ computed in [36] does show the above behaviour[29]. In the star channel computation the Virasoro contribution to the vacuum block was crucial for observing this exponential growth as the global block contribution did not show any exponential growth in the Regge limit. However despite the Virasoro contribution to the comb channel we find no exponential growth for the out of time ordering considered in this subsection. The computation of 6pt correlators when 2 of the pairs are heavy is done without assuming any particular channel with regards to the heavy operators. The only assumption is that the like operators fuse to produce the stress tensor *i.e.* the vacuum block. Therefore in the limit the $\phi$ operators become heavy ($\phi \to \Phi$) in the vacuum block correlator $\langle \phi_{1,2} X_{3,5}, Y_{4,6} \rangle_{\text{vac}} \to \langle \Phi_{1,2} X_{3,5}, Y_{4,6} \rangle_{\text{vac}}$ it is fair to assume that both star and comb channel vacuum blocks may contribute. However what we find is that the 4pt OTO of the light operators $X$ & $Y$ grows only in the star channel and not in the particular comb channel considered here.

### 4.3 OTO for simpler higher pt functions

Higher point correlators can show scrambling for a larger time than that of the 4pt correlators for the case where the correlators are "maximally braided" in terms of their time ordering [21][36]. Since we have shown that reparametrization modes can be used to compute simpler higher point generalizations of lower point correlators in subsection-3.3.1, we can ask whether such higher point correlators can be used to see the larger scrambling times as compared to the lower point correlators from which they were built. We answer this in the simple case of the vacuum block of the 4pt function of 2 like operators generalized to that of the 6pt point

---

[28]Note here we have assumed the domination of the vacuum block.

[29]Although the authors in [36] compute the maximally braided OTOC for the three pairs, one can ascertain the behaviour of 4pt OTO in the 6pt function easily from the branch cuts being crossed. We do not show this explicitly here.

function where each pair is replaced by a triplet. The expression for this is given in the second equation in (3.53)

$$\frac{\langle \phi_{1,2,3} \psi_{4,5,6} \rangle}{\langle \phi_{1,2,3} \rangle \langle \psi_{456} \rangle} = 1 + h_\phi h_\psi \langle \mathcal{B}^{(1)}_{123} \mathcal{B}^{(1)}_{456} \rangle \tag{4.28}$$

We consider an out of time ordering similar to those considered in the case of 4pt. functions and demand that like operators are not out of time ordered. We choose the Euclidean times as

$$\text{OTO}: \quad \tau_1 < \tau_4 < \tau_5 < \tau_2 < \tau_3 < \tau_6 \tag{4.29}$$

and the Lorentzian coordinates to be $t_1 = t_2 = t_3 = 0 = \sigma_4 = \sigma_5 = \sigma_6$, $t_4 = t_5 = t_6 = t$ and $\sigma_1 = \sigma_2 = \sigma_3 = \sigma$. One can readily discern the OTO behaviour by writing out

$$\langle \mathcal{B}^{(1)}_{123} \mathcal{B}^{(1)}_{456} \rangle \sim \langle (\mathcal{B}^{(1)}_{12} + \mathcal{B}^{(1)}_{23} + \mathcal{B}^{(1)}_{13})(\mathcal{B}^{(1)}_{45} + \mathcal{B}^{(1)}_{56} + \mathcal{B}^{(1)}_{46}) \rangle \tag{4.30}$$

and observing from the *out-of-time-ordering* considered here that only the terms

$$\langle \mathcal{B}^{(1)}_{12} \mathcal{B}^{(1)}_{46} \rangle, \ \langle \mathcal{B}^{(1)}_{13} \mathcal{B}^{(1)}_{46} \rangle, \ \langle \mathcal{B}^{(1)}_{13} \mathcal{B}^{(1)}_{56} \rangle, \ \langle \mathcal{B}^{(1)}_{12} \mathcal{B}^{(1)}_{56} \rangle \tag{4.31}$$

in (4.30) contribute to the exponential growth as $t \to \infty$ for $t \gg \beta$ upon mapping the correlator to a thermal background. Therefore the 6pt OTOC is given by

$$\frac{\langle \phi_{1,2,3} \psi_{4,5,6} \rangle_{\text{OTO}}}{\langle \phi_{1,2,3} \rangle \langle \psi_{456} \rangle} = 1 + \frac{48\pi i h_\phi h_\psi}{c} e^{\frac{2\pi}{\beta}(t+\sigma)} \tag{4.32}$$

Note that the scrambling time lasts for the same amount of time $t_* = \frac{\beta}{2\pi} \log c$.

One can repeat the above exercise for the simpler 8pt. and 9pt. correlators considered in section-3 by considering similarly generalizations of the out of time ordering considered for the 6pt. comb channel. We do not pursue this here but note that since the these correlators are evaluated to $1/c^2$ order the scrambling time can at best last for $2t_*$ as was in the case 6pt. comb channel. However it seems to suggest that the generalized 8 and 9 pt. functions of the 6pt function of 3 pairs of like operators would not show scrambling for larger times for the same kind of out of time ordering.

## 5 Bulk perspective

In this section we note certain similarities with the bulk computation demonstrating an exponential growth in OTOC as shown in [18]. We concern ourselves with a static black hole in $AdS_3$. This computation relies on the twin sided eternal Schwarzchild black hole in $AdS_3$ to compute the late time correlation between 2 pairs of operators. The pairs of operators are arranged so as to give an OTOC when suitably analytically continued for time separations much larger than the inverse temperature of the black hole *i.e.*: $t \gg \frac{\beta}{2\pi}$. This would imply that the dual large-$N$ strongly coupled system is being probed with an OTOC with the same operators for time scales much larger than the dispersion time which is of the scale of $\frac{\beta}{2\pi}$.

The particular details of this computation have since been generalized to rotating geometries [35]. The essential idea being that at times $t \gg \frac{\beta}{2\pi}$ the leading order contribution to the probe approximation can be computed from an Eikonal approximation. Here one considers a scattering of shock-waves produced in the bulk by the dual scalar fields interacting by exchanging gravitons governed by the minimal coupling of the scalars in the bulk. In the Kruskal coordinates the Schwarzchild $BTZ$ is

$$\frac{ds^2}{\ell^2} = \frac{-4dudv}{(1+uv)^2} + r_+^2 \frac{(1-uv)^2}{(1+uv)^2} dx^2 \tag{5.1}$$

In response to a in falling shock-wave with momentum $p^v$ produced by a probe scalar sourced at $\phi$ on the boundary at late times

$$T_{uu} = \frac{1}{2r_+} p^v \delta(u) \delta(x - x') \tag{5.2}$$

the above metric (5.1) produces a response [14, 52–55]

$$\frac{ds^2}{\ell^2} = \frac{-4dudv}{(1 + uv)^2} + r_+^2 \frac{(1 - uv)^2}{(1 + uv)^2} dx^2 + h_{uu} du^2$$

$$h_{uu} = 32\pi G_N r_+ p^v \delta(u) g(x) \tag{5.3}$$

This is constrained by the differential equation arising from the terms linear in $G_N$ from the Einstein's equation with stress-tensor (5.2) as $G_N \to 0$.

$$\partial^2 g(x) - r_+^2 g(x) = -\delta(x) \tag{5.4}$$

The eikonal phase shift due to scattering an ingoing shock wave with momentum $p^v$ with an outgoing one with momentum $p^u$ is then given by computing the change in the linearised on-shell action

$$\delta S_{\text{on shell}} = \frac{1}{2} \int d^3 x \sqrt{-g} h_{uu} T^{uu}$$

$$= 4\pi G_N r_+ p^v p^u g(\delta x) \tag{5.5}$$

where $\delta x$ is the difference in the location at the boundary for the sources of the 2 shock waves *i.e.* the location of the pair of operators which source bulk fields.

The CFT understanding of the growth of OTOC thus deduced is that it is governed by the stress-tensor block in the 4pt function. The contribution of other (heavy) operator blocks is ignored by appealing to sparseness of spectrum for holographic systems. Conformal blocks in a 2d CFT are constrained by relevant Casimir equations

$$D\mathcal{F}_{\Delta,l}(u, v) = \lambda_{\Delta,l} \mathcal{F}_{\Delta,l}(u, v) \tag{5.6}$$
$$D = \left(z^2(1 - z)\partial_z^2 - z^2\partial_z\right) + \left(\bar{z}^2(1 - \bar{z})\partial_{\bar{z}}^2 - \bar{z}^2\partial_{\bar{z}}\right)$$
$$\lambda_{\Delta,l} = \frac{1}{2}\Delta(\Delta - 2) + \frac{l^2}{2}, \qquad \Delta = \frac{h + \bar{h}}{2}, l = \frac{h - \bar{h}}{2}$$

while perturbations of dual fields in the bulk are likewise constrained by their bulk *e.o.m.*. For the case at hand the bulk field dual to the CFT stress-tensor is the metric whose response to the shock-wave *i.e.* late time perturbation due to a scalar propagation is governed by linearised Einstein's eq. (5.4).

Assuming a late time behaviour of the stress-tensor conformal block of the form

$$\mathcal{F}(t, x) \approx \frac{e^{\frac{2\pi}{\beta} t}}{c} g(x) \tag{5.7}$$

one can expand (5.6) for late times, knowing that for late Lorentzian times where

$$t_{1,2} = t > 0, \ x_{1,2} = 0, \ t_{3,4} = 0, \ x_{3,4} = x > 0, \qquad \tau_1 > \tau_3 > \tau_2 > \tau_4 \tag{5.8}$$

the out of time ordered cross ratios behave like

$$z \approx -e^{\frac{2\pi}{\beta}(x-t)} \epsilon_{12}^* \epsilon_{34} \quad \bar{z} \approx -e^{-\frac{2\pi}{\beta}(x+t)} \epsilon_{12}^* \epsilon_{34} \tag{5.9}$$

with $\epsilon_{ij} = i\left(e^{\frac{2\pi}{\beta}\tau_i} - e^{\frac{2\pi}{\beta}\tau_j}\right)$. This late time expansion ($t \gg \frac{\beta}{2\pi}$) of the Casimir equation yields

$$\partial^2 g(x) - r_+^2 g(x) = 0 \tag{5.10}$$

which is precisely the linearised Einstein's eq. (5.4) which we were required to solve for a shock-wave but without the source delta function on the *r.h.s.* Note that it was crucial that we assumed a growing behaviour of the form (5.7) in $t$ for the stress-tensor which can only be assumed to hold for out of time ordering of the 4pt correlators.

Given the fact that the vacuum conformal block at $\mathcal{O}(1/c)$ is given by $\langle \mathcal{B}_{12}^{(1)} \mathcal{B}_{34}^{(1)} \rangle$ *i.e.* two-point functions of bi-locals constructed out of the reparametrization modes $\epsilon_i$s, there seem to be a plausible relation between them and the backreactions in the bulk of the form (5.4). It is worth asking what these reparametrization modes mean in terms of bulk fields and can they similarly furnish a effective description of chaotic degrees of freedom as they do in the CFT.

## 6 Discussion and Conclusions

In this article, we initiated a study of Ward identities for the reparametrization modes in a 2d CFT, in particular we looked at 2 insertions of the $\epsilon$ modes in 2pt and 3pt functions of primaries. We find that just as the expression for Ward identity associated with a single insertion of $\epsilon$ in 2pt function $\langle \epsilon_0 \phi_1 \phi_2 \rangle$ allows one to compute the first subleading correction to the stress-tensor block of 4pt operators $\langle \phi_1 \phi_2 \psi_3 \psi_4 \rangle$, double insertions of $\epsilon$ of the form $\langle \epsilon_{3'} \epsilon_{4'} \phi_1 \phi_2 \rangle$ allows us to compute the Virasoro contribution to the stress-tensor 6pt comb channel as depicted in Figure-1. We also find that such a method of computing the Virasoro contribution to the comb channel can be easily generalized to the case where the identical pairs in the comb channel are changed to identical triplets. In other words the Ward identity for $\langle \epsilon_{3'} \epsilon_{4'} \phi_1 \phi_2 \rangle$ allows us to simpler higher point generalization of the form depicted in Figures -2 & 3.

We also study various out of time ordered behaviour of the Virasoro contribution to the 6pt stress-tensor comb channel after mapping them to a thermal background. It was shown in [36] that even the global part of this correlator grows exponentially like the star vacuum channel 6pt correlator for maximally braided OTO configuration. We find this to be true even for the full Virasoro contribution as expected. We also notice that when only the $X$ and $Y$ pairs are *out-of-time-ordered* (Figure-1) the global part of the answer does not show an exponential growth. This can be thought due to only the global states associated with the $\phi$ operator propagating in the internal leg Figure-1(a), as the global states would correspond to no gravitons propagating in the holographic bulk analogue. Inclusion of the Virasoro contributions in this leg would likewise imply propagation of gravitons. However we find that the inclusion of the Virasoro contribution also does not lead to an exponential growth for this particular *out-of-time-ordering*. We also find that although various branch cuts are crossed the final OTO result does not grow exponentially in time. This we find in contrast with the 6pt star channel analysis in [36] where the inclusion of the Virasoro contribution was necessary for the exponentially growing behaviour.

While this approach provides us with an alternative method of computing higher point correlators, in the stress-tensor dominated channel, it leaves open avenues for investigating further physical aspects. As was emphasised in [36] that the physics of multiple linear graviton exchanges is more important than that of graviton self interaction [56–58], it would be interesting to understand how repeated use of the reparametrization Ward identity of the form (3.18) could help us understand aspects of higher point correlators. Such a program would however require knowing the results more conformal integrals than those are currently available in literature.

The observations of section 5 indicate a plausible relationship between the reparametrization modes and the bulk backreactions to matter fields *via* Einstein's equation. From a holographic perspective it is nonetheless important to understand the bulk analogue for the reparametrization modes as these may similarly capture effective degrees of freedom which encapsulate chaotic behaviour. Such effective descriptions al-

ready exist in terms of the well studied Jackiw-Teitelboim (JT) model [32, 33] used to understand the near horizon dynamics of near extremal black holes. This model captures thermal chaotic behaviour in terms of an effective 1d theory of time reparametrizations in terms of its Schwarzian derivatives at the near horizon throat boundary. However, the phenomenon of extremal chaos as deduced by the results of [34, 35] is not captured by this model [59]. Investigations into the holographic dual description of reparametrization modes and their Ward identities would perhaps yield a more complete picture as they necessarily must reduce to the 1d reparametrization modes of the JT model in the case of near horizon dynamics of near extremal black holes.

At a technical level, it is interesting to understand in detail how this approach may work when the pairwise identical operators are relaxed to a more general configuration of operators, including spin. Spinning operators are particularly important in the understanding the physics of Kerr black holes, see *e.g.* [35, 59, 60] for discussions related to the chaos-bound in this case. An involved and physically interesting description is likely to exist in higher dimensions, for generic operators in the dual CFT [61, 62]. From a Holographic perspective, this is tied to the near horizon physics of rotating black holes, in which a complete understanding is lacking at present [59]. We hope to address some of these questions in near future.

It would also be interesting to understand how this approach can be used in the study of the stress-tensor block for 2 heavy operators ($H \sim c$) inserted along with many light operators $L \ll c$. It was shown for HHLL [26] that this is obtained from a conformal transformation of the LLLL with the transformation parameter governed by $\sqrt{1 - 24H/c}$. A similar but stronger statement was proved using the monodromy method to some extent for the case of HHLLLL.... case in [44] where in it was argued that even sub-leading corrections in $1/c$ can be obtained by employing a similar change of conformal frame. Although this does not match the expected answer for the 6pt vacuum block at $\mathcal{O}(c^{-2})$ in the limit the heavy operators tend to being light. A more clear understanding in this regard would shed further light on the Eigenstate Thermalization Hypothesis in $CFT_2$.

On a more conceptual note, recent advances in understanding the properties of thermal correlators, including that of OTOCs, makes it clear that the IR-physics encoded in these correlators implicitly know about the UV-completion, specially for systems with a Holographic dual. This statement simply follows from e.g.the chaos bound, which is inherently related to unitarity in the high energy states, that nonetheless provides a bound for an IR-quantity, *i.e.* the Lyapunov exponent. A more general understanding of this aspect is still missing, and it is a very interesting question to what extent the reparametrization modes, together with the shadow operator formalism, Ward identities and such, can shed light on such aspects.

On a related note, it is curious that the dynamics of maximal chaos, in Holography, does not necessarily require an Einstein-Hilbert dynamics. Instead, similar physics can be obtained from a Nambu-Goto dynamics[63, 64] or a Dirac-Born-Infeld dynamics[65]. In the former case, with strings propagating in an AdS$_3$-background, there is a precise relation between the dual CFT and the world-sheet CFT[66]. It would be very interesting to uncover the details of how the reparametrization modes of these two CFTs are related to each other. We hope to come back to some of these issues in near future.

## Acknowledgements

The authors would like to thank Bobby Ezhuthachan for useful comments on an earlier version of this manuscript. The authors are also grateful to the anonymous referees of Sci-Post to have insisted on necessary checks of the claims made in an earlier version of this manuscript. AKP is supported by the Council of Scientific & Industrial Research (CSIR) Fellowship No. 09/489(0108)/2017-EMR-I. RP is supported by the Lise Meitner fellowship M 2882-N funded by the FWF.

## A  Embedding space

The embedding space of $d + 2$ dimensions allows the realization of the conformal group as $SO(d + 1, 1)$ rotation. The space-time coordinates are obtained by projecting onto the null sphere in the embedding space and identifying scaling $w.r.t.$ the affine parameter on the null sphere. The null sphere given by

$$X^a \cdot X_a = X^+ X^- + X^\mu X_\mu = 0 \tag{A.1}$$

can be used to set

$$X^- = -X^+ x^2, \quad X^\mu = X^+ x^\mu \text{ for } X^+ \neq 0$$
$$X^a = X^+ \{1, -x^2, x^\mu\} \tag{A.2}$$

Using this parametrization of the null sphere we see that $X^+$ is the affine parameter. Further we identify $X^+ \equiv \lambda X^+, \forall \lambda \in \mathbb{R}$. The projector onto the null surface and normal vectors are obtained by

$$e_\mu^a(X) = \frac{\partial X^a}{\partial x^\mu} = X^+ \{0, -2x^\mu, \delta_\mu^\nu\},$$

$$k^a = \frac{\partial X^a}{\partial X^+} = \{1, -x^2, x^\mu\}$$

$$N^a = 2\delta_-^a, \tag{A.3}$$

where $N^a$ is obtained by demanding $k \cdot N = 1$ & $e_\mu \cdot N = 0$. The space-time fields are obtained from the embedding space fields by restricting them to the null sphere. Space-time primary scalar $\phi(x)$ is given by

$$\Phi(X) \equiv X^+ \phi(x) \tag{A.4}$$

where $\Phi(X)$ is restricted on the null sphere. Similarly for space-time tensor primaries we have

$$V_{\mu_1 \ldots \mu_l}(x) \equiv e_{\mu_1}^{a_1}(X) \ldots e_{\mu_l}^{a_l}(X) V_{a_1 \ldots a_l}(X). \tag{A.5}$$

where $\equiv$ is understood as having to identify (gauge fix) $X^+$ components (to 1). One can then define the inversion tensor in embedding space as

$$I_{ab}(X_1^a, X_2^b) = \eta_{ab} - \frac{X_b^1 X_a^2}{X_{12}}, \quad \text{where} \quad X_{12} = X_1 \cdot X_2 = -\tfrac{1}{2} X_1^+ X_2^+ x_{12}^2,$$

$$\text{as} \quad e_\mu^a(X_1) I_{ab} e_\nu^b(X_2) = I_{\mu\nu} = \eta_{\mu\nu} - 2 \frac{x_\mu^{12} x_\nu^{12}}{x_{12}^2}. \tag{A.6}$$

Here we have used $e_\mu(X_1) \cdot X_2 = -X_1^+ X_2^+ x_\mu^{12}$. The embedding space metric can therefore be decomposed along the infinitesimal curves (A.3) as

$$\eta_{ab} = e_\mu^a e_\nu^b \eta^{\mu\nu} + 2k^{(a} N^{b)}. \tag{A.7}$$

Tensor primaries in embedding space would also satisfy

$$V^a(X) \cdot X_a = 0 \implies V^a(X) \equiv V^a(X) + X^a s(X) \tag{A.8}$$

on the null surface $X^2 = 0$. One can then show using (A.8) and (A.7) that

$$e_\mu^a(X) I_{ab}(X, Y) V^b(Y) \equiv I_{\mu\nu}(x, y) V^\nu(y) \tag{A.9}$$

This would be useful in defining shadows of tensor primaries in terms of their space-time components.

## B Conformal Integrals

We note certain useful results for conformal integrals in $d$ and $d = 2$ dimensions here [29, 30]. We indicate the $d$ dimensional conformally invariant volume in the $d + 2$ dimensional embedding space as $D^d X$ here but revert to using $d^d X$ or $d^d x$ in the main text while treating them as conformally invariant in $d$ dimensions.

$$I(Y) = \int D^d X \frac{1}{(-2X.Y)^d} = \frac{\pi^{d/2} \Gamma(d/2)}{\Gamma(d)} \frac{1}{(Y^2)^{d/2}}, \qquad (\forall \, Y^2 < 0) \tag{B.1}$$

$$\int D^d X_0 \frac{1}{X_{10}^a X_{02}^b X_{03}^c} = \frac{\pi^{\frac{d}{2}} \Gamma(\frac{d}{2} - a)\Gamma(\frac{d}{2} - b)\Gamma(\frac{d}{2} - c)}{\Gamma(a)\Gamma(b)\Gamma(c)} \frac{1}{X_{12}^{\frac{d}{2}-c} X_{13}^{\frac{d}{2}-b} X_{23}^{\frac{d}{2}-a}} \tag{B.2}$$

where $a + b + c = d$ and $X_{ij} = -2X_i.X_j = (x_i - x_j)^2$ when $X_i^2 = X_j^2 = 0$.

$$\int \frac{D^d X_0}{X_{10}^{d-\Delta} X_{20}^{\Delta}} = \frac{\pi^{\frac{d}{2}} \Gamma(\Delta - \frac{d}{2})}{\Gamma(\Delta)} \frac{(X_2^2)^{\frac{d}{2}-\Delta}}{X_{12}^{d-\Delta}} \tag{B.3}$$

Using this one can show

$$\int \frac{D^d X_0 D^d X_1}{X_{10}^{d-\Delta} X_{20}^{\Delta} X_{13}^{\Delta}} = \frac{\pi^{\frac{d}{2}} \Gamma(\Delta - \frac{d}{2})\Gamma(\frac{d}{2} - \Delta)}{\Gamma(\Delta)\Gamma(d - \Delta)} \frac{1}{X_{23}^{\Delta}} \tag{B.4}$$

In the 2d case we use the integrals of he form

$$I_n = \frac{1}{\pi} \int d^2 x_0 \, f_n(z_0)\bar{f}_n(\bar{z}_0), \qquad f_n(z_0) = \prod_{i=1}^{n} (z_0 - z_i)^{-h_i}, \ \bar{f}_n(\bar{z}_0) = \prod_{i=1}^{n} (\bar{z}_0 - \bar{z}_i)^{-\bar{h}_i}$$

$$\text{where} \qquad \sum_{i=1}^{n} h_i = \sum_{i=1}^{n} \bar{h}_i = d = 2, \quad h_i - \bar{h}_i \in \mathbb{Z}. \tag{B.5}$$

These integrals were solved for $n = 2, 3, 4$ cases in [30](Appendix A). We note the $n = 4$ case for our use below

$$I_4 = z_{12}^{h_3+h_4-1} z_{23}^{h-1+h_4-1} z_{31}^{h_2-1} z_{24}^{-h_4} \bar{z}_{12}^{\bar{h}_3+\bar{h}_4-1} \bar{z}_{23}^{\bar{h}-1+\bar{h}_4-1} \bar{z}_{31}^{\bar{h}_2-1} \bar{z}_{24}^{-\bar{h}_4} \mathcal{I}_4(z,\bar{z}),$$

$$\mathcal{I}_4 = K_4 \, {}_2F_1(1 - h_2, h_4; h_3 + h_4, z) \, {}_2F_1(1 - \bar{h}_2, \bar{h}_4; \bar{h}_3 + \bar{h}_4, \bar{z})$$
$$+ \bar{K}_4 (-1)^{h_1+h_4-\bar{h}_1-\bar{h}_4} z^{h_1+h_2-1} \bar{z}^{\bar{h}_1+\bar{h}_2-1} \, {}_2F_1(1 - h_3, h_1; h_1 + h_2, z)$$
$$\times {}_2F_1(1 - \bar{h}_3, \bar{h}_1; \bar{h}_1 + \bar{h}_2, \bar{z}),$$

$$K_4 = \frac{\Gamma(1 - h_1)\Gamma(1 - h_2)\Gamma(h_1 + h_2 - 1)}{\Gamma(\bar{h}_1)\Gamma(\bar{h}_2)\Gamma(2 - \bar{h}_1 - \bar{h}_2)}, \quad \bar{K}_4 = \frac{\Gamma(1 - h_3)\Gamma(1 - h_4)\Gamma(h_3 + h_4 - 1)}{\Gamma(\bar{h}_3)\Gamma(\bar{h}_4)\Gamma(2 - \bar{h}_3 - \bar{h}_4)} \tag{B.6}$$

where $z = \frac{z_{12} z_{34}}{z_{13} z_{24}}, \ \bar{z} = \frac{\bar{z}_{12} \bar{z}_{34}}{\bar{z}_{13} \bar{z}_{24}}$

## C $\epsilon$-mode Ward Identity *via* integration

In this Appendix we evaluate the $\langle \epsilon_3 \epsilon_4 \phi_1 \phi_2 \rangle$ using integrating the Ward identity namely

$$\langle T_{-1} T_0 \phi_1 \phi_2 \rangle = \frac{c/2}{z_{-10}^4} \langle \phi_1 \phi_2 \rangle + \left( \frac{(h\text{-}1)z_{12}^2}{z_{-11}^2 z_{-12}^2} + \frac{z_{01}^2}{z_{-10}^2 z_{-11}^2} + \frac{z_{02}^2}{z_{-10}^2 z_{-12}^2} \right) \frac{h z_{12}^2}{z_{01}^2 z_{02}^2} \frac{1}{(z_{12}\bar{z}_{12})^{2h}}. \tag{C.1}$$

and then expressing the result as a total derivative *wrt* $\bar{z}_{3,4}$. Consider the integral of the first term inside the brackets above[30]:

$$\int_{-1,0} \frac{z_{-13}^2 z_{40}^2}{\bar{z}_{-13}^2 \bar{z}_{40}^2} \frac{z_{12}^2}{z_{-11}^2 z_{-12}^2} \frac{z_{12}^2}{z_{01}^2 z_{02}^2} \frac{1}{(z_{12}\bar{z}_{12})^{2h}} = \int_{-1,0} \frac{z_{-13}^4 z_{40}^4 z_{12}^4 \bar{z}_{-12}^2 \bar{z}_{01}^2 \bar{z}_{02}^2}{X_{12}^{2h} X_{-13}^2 X_{40}^2 X_{01}^2 X_{02}^2 X_{-11}^2 X_{-12}^2}$$

---

[30] We consider the all $z$ components of the resulting tensor.

$$= \frac{1}{X_{12}^{2h-2}} \int_0 \frac{I_{40}^{a\bar{a}} I_{40}^{b\bar{b}} I_{\bar{a}\bar{a}}^{01} I_{\bar{b}\bar{b}}^{02} I_{12}^{\tilde{a}\tilde{b}}}{X_{40}^0 X_{01} X_{02}} \int_{-1} \frac{I_{a\bar{a}}^{3\text{-}1} I_{b\bar{b}}^{3\text{-}1} I_{\bar{a}\bar{a}}^{\text{-}11} I_{\bar{b}\bar{b}}^{\text{-}12} I_{12}^{\tilde{a}\tilde{b}}}{X_{\text{-}13}^0 X_{\text{-}11} X_{\text{-}12}} \Bigg|_{zzzz} \tag{C.2}$$

where we evaluate the all $z$ component of the last expression. Note that each integral is similar to the integral used for obtaining $\langle \tilde{B}_3^{ab} \phi_1 \phi_2 \rangle$ from $\langle B_{ab}^0 \phi_1 \phi_2 \rangle$ with the dimension of $\tilde{B}_{ab} = \delta \to 0$.

$$\frac{1}{X_{12}^{2h}} \int_{-10} \frac{z_{\text{-}13}^2 z_{40}^2}{\bar{z}_{\text{-}13}^2 \bar{z}_{40}^2} \frac{z_{12}^2}{z_{\text{-}11}^2 z_{\text{-}12}^2} \frac{z_{12}^2}{z_{01}^2 z_{02}^2} = \frac{4}{X_{12}^{2h}} \frac{z_{13} z_{14} z_{23} z_{24} \bar{z}_{12}^2}{\bar{z}_{13} \bar{z}_{14} \bar{z}_{23} \bar{z}_{24} z_{12}^2}$$
$$= \frac{4}{X_{12}^{2h}} \left( I_{41}^{a\bar{a}} I_{\bar{a}\bar{b}}^{12} I_{24}^{\bar{b}b} \right) \left( I_{31}^{c\bar{c}} I_{\bar{c}\bar{d}}^{12} I_{23}^{\bar{d}d} \right) \Bigg|_{zzzz}$$
$$= \left( \frac{\pi c}{6} \right)^2 \frac{4\bar{\partial}_4 \bar{\partial}_3}{X_{12}^{2h}} \langle \epsilon_3 \mathcal{B}_{12}^{(1)} \rangle \langle \epsilon_4 \mathcal{B}_{12}^{(1)} \rangle \tag{C.3}$$

In going from the second line to the third line we note the equivalence between the $rhs$s of (3.3) and (3.4). The integral for the second term in side the brackets in (C.1) can be written as

$$\int_{-1,0} \frac{z_{40}^2 z_{\text{-}13}^2}{\bar{z}_{40}^2 \bar{z}_{\text{-}13}^2} \frac{z_{12}^2}{z_{\text{-}10}^2 z_{\text{-}11}^2 z_{02}^2 X_{12}^{2h}} = \int_{-1,0} \frac{z_{40}^4 z_{3\text{-}1}^4 z_{12}^2 \bar{z}_{\text{-}10}^2 \bar{z}_{\text{-}11}^2 \bar{z}_{02}^2}{X_{40}^2 X_{\text{-}13}^2 X_{\text{-}10}^2 X_{\text{-}11}^2 X_{02}^2 X_{12}^{2h}}$$
$$= \int_{-1,0} \frac{I_{40}^{a\bar{a}} I_{40}^{b\bar{b}} I_{\bar{b}e}^{02} I_{21}^{ef} I_{\bar{c}f}^{\text{-}11} I_{\bar{d}\bar{a}}^{\text{-}10} I_{3\text{-}1}^{d\bar{d}} I_{\text{-}13}^{c\bar{c}}}{X_{40}^0 X_{\text{-}13}^0 X_{\text{-}10} X_{\text{-}11} X_{02} X_{12}^{2h-1}} \Bigg|_{zzzz} . \tag{C.4}$$

The third term inside the brackets in (C.1) is obtained by exchanging $X_1 \leftrightarrow X_2$. The above integral unlike (C.2) does not take a familiar form. However such integrals where explicitly known in 2d $c.f.$ appendix A of [30]. For the case at hand we note the relevant integral in Appendix B here.

$$\int_{-1,0} \frac{z_{40}^2 z_{\text{-}13}^2}{\bar{z}_{40}^2 \bar{z}_{\text{-}13}^2} \frac{z_{12}^2}{z_{\text{-}10}^2 z_{\text{-}11}^2 z_{02}^2 X_{12}^{2h}} = \frac{\bar{\partial}_4 \bar{\partial}_3}{X_{12}^{2h}} \left\{ 6 \frac{z_{13}^2 z_{24}^2}{z_{12}^2} \left( \log(1-z) + \text{Li}_2(\bar{z}) \right) - z_{34}^2 \log(\bar{z}_{34}) \right.$$
$$\left. + \frac{z_{13} z_{24} z_{34}}{z_{12}} \left( 6 \log(\bar{z}_{34}) - 4 \log(1-z) \log(\bar{z}_{34}) - 4\text{Li}_2(\bar{z}) \right) \right\} \tag{C.5}$$

We can write the $l.h.s$ above as the all $z$ component of

$$\frac{\partial_4^a \partial_3^c}{X_{12}^{2h}} \left\{ 6 \frac{X_{13} X_{24}}{X_{12}} I_{31}^{b\bar{b}} I_{\bar{b}\bar{d}}^{12} I_{24}^{\bar{d}d} \left( \log(1-z) + \text{Li}_2(\bar{z}) \right) - I_{34}^{bd} X_{34} \log(X_{34}) \right.$$
$$\left. + \frac{X_{34}}{\bar{z}} I_{31}^{b\bar{b}} I_{\bar{b}\bar{d}}^{12} I_{24}^{\bar{d}d} \left( 6 \log(X_{34}) - 4 \log(1-z) \log(X_{34}) - 4\text{Li}_2(\bar{z}) \right) \right\} \tag{C.6}$$

where $z = \frac{z_{12} z_{34}}{z_{13} z_{24}}$ and it's complex conjugate are related to the conformally invariant cross ratios as $u = z\bar{z}$, $v = (1-z)(1-\bar{z})$. Therefore we can now write out $\langle \epsilon \epsilon \phi \phi \rangle$ as

$$\frac{\langle \epsilon_3 \epsilon_4 \phi_1 \phi_2 \rangle}{\langle \phi_1 \phi_2 \rangle} = \langle \epsilon_3 \epsilon_4 \rangle + h(h-1) \langle \epsilon_3 \mathcal{B}_{12}^{(1)} \rangle \langle \epsilon_4 \mathcal{B}_{12}^{(1)} \rangle +$$
$$+ h \left( \frac{12}{c} \right)^2 \left\{ 6 \frac{X_{13} X_{24}}{X_{12}} I_{31}^{b\bar{b}} I_{\bar{b}\bar{d}}^{12} I_{24}^{\bar{d}d} \left( \log(1-z) + \text{Li}_2(\bar{z}) \right) - I_{34}^{bd} X_{34} \log(X_{34}) \right.$$
$$\left. + \frac{X_{34}}{\bar{z}} I_{31}^{b\bar{b}} I_{\bar{b}\bar{d}}^{12} I_{24}^{\bar{d}d} \left( 6 \log(X_{34}) - 4 \log(1-z) \log(X_{34}) - 4\text{Li}_2(\bar{z}) \right) \right.$$
$$\left. + (1 \leftrightarrow 2) \right\}_{zz} \tag{C.7}$$

Restricting to only the physical block and simplifying the result in terms of the cross ratio we find

$$\frac{\langle \epsilon_3 \epsilon_4 \phi_1 \phi_2 \rangle_{\text{phys}}}{\langle \phi_1 \phi_2 \rangle} = \langle \epsilon_3 \epsilon_4 \rangle_{\text{phys}} + h(h-1) \langle \epsilon_3 \mathcal{B}_{12}^{(1)} \rangle_{\text{phys}} \langle \epsilon_4 \mathcal{B}_{12}^{(1)} \rangle_{\text{phys}} + h \left( \frac{12}{c} \right)^2 \mathcal{C}_{\text{phys}}^{(2)}$$

$$\text{(C.8)}$$

where

$$\mathcal{C}^{(2)}_{\text{phys}} = \langle \epsilon_3 \epsilon_4 \rangle_{\text{phys}} \left[ 4 + \left( -2 + \frac{4}{z} \right) \log(1-z) \right] + z_{34}^2 \mathcal{F}(z). \tag{C.9}$$

The extra term $\mathcal{F}(z)$ is undetermined and constraints of the form (2.17) can be further used to determine it.

## D   Symmetrization

In this appendix we compute the connected contribution to

$$V^{(6)Vir}_{T,comb} = \frac{\langle X_3 X_5 \,|\mathbb{I}|\, \phi_1 \; \phi_2 \,|\mathbb{I}|\, Y_4 Y_6 \rangle}{\langle \phi_1 \phi_2 \rangle \langle X_3 X_5 \rangle \langle Y_4 Y_6 \rangle} \tag{D.1}$$

upto $\mathcal{O}(1/c^2)$. Here $\mathbb{I}$ denote the projectors onto the vacuum to the required order in $1/c$. Thus we have

$$\mathbb{I} = 1 + I_1 + I_2 + \dots$$

$$I_1 = \sum_i \mathcal{N}^{-1}_{i,i} \; L_{-i} \rangle \langle L_i$$

$$I_2 = \sum_{m,n} L_{-(m,n)} \rangle \left[ \mathcal{N}^{-1}_{(m,n),(m,n)} \langle L_{(m,n)} + \mathcal{N}^{-1}_{(m,n),(m+n)} \langle L_{m+n} \right] + \mathcal{N}^{-1}_{(m,n),(m+n)} L_{-(m+n)} \rangle \langle L_{(m,n)}$$

$$+ \mathcal{N}^{-1}_{(m+n),(m+n)} \; L_{-(m+n)} \rangle \langle L_{(m+n)} \tag{D.2}$$

where $\mathcal{N}^{-1}_{i,i}$ in $I_1$ goes as $1/c$ while the $\mathcal{N}^{-1}$s in $I_2$ all go as $1/c^2$. We would only need terms upto $I_2$ in the $1/c$ expansion of $\mathbb{I}$. $V^{(6)Vir}_{T,comb}$ can be expanded upto $\mathcal{O}(c^{-2})$ as

$$\begin{aligned}
V^{(6)Vir}_{T,comb} = \frac{\langle X_{3,5}|\mathbb{I}|\phi_{1,2}|\mathbb{I}|Y_{4,6}\rangle}{\langle X_{3,5}\rangle\langle\phi_{1,2}\rangle\langle Y_{4,6}\rangle} &= 1 + \frac{\langle X_{3,5}\rangle\langle\phi_{1,2}|I_1|Y_{4,6}\rangle}{\langle X_{3,5}\rangle\langle\phi_{1,2}\rangle\langle Y_{4,6}\rangle} + \frac{\langle X_{3,5}|I_1|\phi_{1,2}\rangle\langle Y_{4,6}\rangle}{\langle X_{3,5}\rangle\langle\phi_{1,2}\rangle\langle Y_{4,6}\rangle} \\
&+ \frac{\langle X_{3,5}\rangle\langle\phi_{1,2}|I_2|Y_{4,6}\rangle}{\langle X_{3,5}\rangle\langle\phi_{1,2}\rangle\langle Y_{4,6}\rangle} + \frac{\langle X_{3,5}|I_2|\phi_{1,2}\rangle\langle Y_{4,6}\rangle}{\langle X_{3,5}\rangle\langle\phi_{1,2}\rangle\langle Y_{4,6}\rangle} + \frac{\langle X_{3,5}|I_1|\phi_{1,2}|I_1|Y_{4,6}\rangle}{\langle X_{3,5}\rangle\langle\phi_{1,2}\rangle\langle Y_{4,6}\rangle} \\
&+ \frac{\langle X_{3,5}|I_1|\phi_{1,2}|I_2|Y_{4,6}\rangle}{\langle X_{3,5}\rangle\langle\phi_{1,2}\rangle\langle Y_{4,6}\rangle} + \frac{\langle X_{3,5}|I_2|\phi_{1,2}|I_1|Y_{4,6}\rangle}{\langle X_{3,5}\rangle\langle\phi_{1,2}\rangle\langle Y_{4,6}\rangle} \\
&+ \frac{\langle X_{3,5}|I_2|\phi_{1,2}|I_2|Y_{4,6}\rangle}{\langle X_{3,5}\rangle\langle\phi_{1,2}\rangle\langle Y_{4,6}\rangle} + \dots
\end{aligned} \tag{D.3}$$

Here we have retained only those terms that could possibly contribute at $\mathcal{O}(c^{-2})$. It is worth noticing the structure in the $1/c$ expansion above. The first line in the *rhs* goes as $1/c$. The first 2 terms in the second line contribute at $1/c^2$ but are disconnected. The third term in the second line contributes non-trivially at $1/c^2$ and has been computed in $\widehat{V}^{(6)Vir}_{T,comb}$, however this term also contributes at order $1/c$ when the $L$s acting on $\phi_{1,2}$ contract with each other[31]. Tis contribution is exactly

$$\frac{\langle\phi_{1,2}\rangle\langle X_{3,5}|I_1|Y_{4,6}\rangle}{\langle X_{3,5}\rangle\langle\phi_{1,2}\rangle\langle Y_{4,6}\rangle} \sim \mathcal{O}(1/c) \tag{D.4}$$

Notice this term along with the terms in the first line of the *rhs* effectively makes the $1/c$ contribution symmetric *wrt* the 3 pairs of operators. We shall see this happen order by order in the $1/c$ expansion.

---

[31]Note that $\widetilde{V}^{(6)Vir}_{T,comb}$ computes the $1/c^2$ contribution to $\frac{\langle X_{3,5}|I_1|\phi_{1,2}|I_1|Y_{4,6}\rangle}{\langle X_{3,5}\rangle\langle\phi_{1,2}\rangle\langle Y_{4,6}\rangle}$ where the stress-tensors acting on $\phi_{1,2}$ do not contract with each other. This is denoted with a prime.

Consider the term in the 4th line of the *rhs* in (D.3) which gives a non-trivial contribution at $\mathcal{O}(1/c^2)$

$$\langle X_{3,5}|I_2|\phi_{1,2}|I_2|Y_{4,6}\rangle = \sum_{m,n,a,b} \langle X_{3,5}| \left[ \mathcal{N}^{-1}_{(m,n),(m,n)} L_{-(m,n)}\rangle\langle L_{(m,n)} + \ldots \right] |\phi_{1,2}| \left[ \mathcal{N}^{-1}_{(a,b),(a,b)} L_{-(a,b)}\rangle\langle L_{(a,b)} + \ldots \right] |Y_{4,6}\rangle \tag{D.5}$$

where the dots denote the other terms in $I_2$. The $\mathcal{N}^{-1}$s in the rhs above contribute a total of $1/c^4$, therefore we need to consider contractions between $L_{(m,n)}$ and $L_{-a,b}$ in $\langle L_{(m,n)}\phi_{3,5}L_{-a,b}\rangle \sim c^2$ occurring in the rhs above. All other possibilities are $\mathcal{O}(1/c^3)$ or lower. This contraction basically implies

$$\langle X_{3,5}|I_2|\phi_{1,2}|I_2|Y_{4,6}\rangle = \langle X_{3,5}|I_2|Y_{4,6}\rangle\langle\phi_{1,2}\rangle + \mathcal{O}(1/c^3) \sim \mathcal{O}(1/c^2) \tag{D.6}$$

The first term in the *rhs* above taken together with the first 2 terms in the second line of the *rhs* in (D.3) symmetrizes the disconnected contribution at order $\mathcal{O}(c^{-2})$.

Consider next one of the terms in the third line in the *rhs* in (D.3) which contributes non-trivially at $\mathcal{O}(c^{-3})$

$$\langle X_{3,5}|I_2|\phi_{1,2}|I_1|Y_{4,6}\rangle = \sum_{m,n,i} \langle X_{3,5}| \left[ \mathcal{N}^{-1}_{(m,n),(m,n)} L_{-(m,n)}\rangle\langle L_{(m,n)} + \ldots \right] |\phi_{1,2}|\mathcal{N}^{-1}_{i,i} \ L_{-i}\rangle\langle L_i|Y_{4,6}\rangle \tag{D.7}$$

Here we have to consider contractions between the generators in $I_1$ and $I_2$ as otherwise the contribution from $\mathcal{N}^{-1}$s would be $\mathcal{O}(1/c^3)$. Here we are interested in only those contractions which result in connected diagrams, therefore we are concerned with terms where $L_{-(m,n)}$ acts on $X_{3,5}$ and $L_{(m,n)}$ on $\phi_{1,2}$. There are 2 possible contractions of $L$s; the one in which $L_{-i}$ contracts with all the generators in $L_{(m,n)}$ in the box bracket above give zero as the expression

$$\left[ \mathcal{N}^{-1}_{(m,n),(m,n)} \langle L_{(m,n)}L_{-i}\rangle + \mathcal{N}^{-1}_{(m,n),(m+n)} \langle L_{m+n}L_{-i}\rangle \right] = 0 \tag{D.8}$$

vanishes identically. The only surviving term in (D.7) is obtained when $L_{-i}$ contracts with only one of the generators in $L_{(m,n)}$ resulting in

$$\begin{aligned}
\langle X_{3,5}|I_2|\phi_{1,2}|I_1|Y_{4,6}\rangle &= \sum_{m,n} \mathcal{N}^{-1}_{(m,n),(m,n)} \langle X_{3,5}L_{-(m,n)}\rangle\langle\phi_{1,2}L_{-m}\rangle\langle Y_{4,6}L_{-n}\rangle + \mathcal{O}(1/c^3) \\
&= \sum_{m,n} \mathcal{N}^{-1}_{(m,n),(m,n)} \langle\phi_{1,2}L_{-n}\rangle\langle L_n X_{3,5}L_{-m}\rangle\langle L_m Y_{4,6}\rangle + \mathcal{O}(1/c^3) \\
&= \langle\phi_{1,2}|I_1|X_{3,5}|I_1|Y_{4,6}\rangle' + \mathcal{O}(1/c^3) \sim \mathcal{O}(1/c^2) \tag{D.9}
\end{aligned}$$

where the prime denotes that the term has no further contribution from contractions of $L$s. In other words the above contraction exchanges $\phi_{1,2} \leftrightarrow X_{3,5}$ in $\widetilde{V}^{(6)Vir}_{T,comb}$. Similarly

$$\langle X_{3,5}|I_1|\phi_{1,2}|I_2|Y_{4,6}\rangle = \langle\phi_{1,2}|I_1|Y_{4,6}|I_1|X_{3,5}\rangle' + \mathcal{O}(1/c^3) \sim \mathcal{O}(1/c^2) \tag{D.10}$$

Therefore collecting the contributions at order $1/c^2$ from the third term in the second line and the terms from the third line in (D.3) we have have the connected 6pt Virasoro contribution to the stress-tensor comb channel as

$$V^{(6)\text{Vir}}_{T,\text{comb}} = h_X h_Y \frac{\langle\mathcal{B}^{(1)}_{35}\mathcal{B}^{(1)}_{46}\phi_1\phi_2\rangle'}{\langle\phi_1\phi_2\rangle} + h_\phi h_Y \frac{\langle\mathcal{B}^{(1)}_{12}\mathcal{B}^{(1)}_{46}X_3X_5\rangle'}{\langle X_3X_5\rangle} + h_\phi h_X \frac{\langle\mathcal{B}^{(1)}_{35}\mathcal{B}^{(1)}_{12}Y_4Y_6\rangle'}{\langle Y_4Y_6\rangle} + \mathcal{O}(1/c^3) \tag{D.11}$$

*Note:* the above set of arguments can be repeated as it is when any of the pairs of $\phi$s, $X$s and $Y$s are changed to triplets as in the case of the 8pt. and 9pt. functions dealt in section-3.3.1.

# E  Virasoro stress-tensor comb channel for simple 8pt and 9pt functions.

Here we explicitly state the results for the Virasoro stress-tensor comb channels of simpler higher point generalizations of pair-wise equal 6pt function of light operators *i.e.* $\langle X_3 X_5 X_7 \phi_1 \phi_2 Y_4 Y_6 Y_8 \rangle$ as in Figure-2(b) and $\langle X_3 X_5 X_7 \phi_0 \phi_1 \phi_2 Y_4 Y_6 Y_8 \rangle$ in Figure-3(b). These also occur at $\mathcal{O}(1/c^2)$. The only expressions we need are the 4pt vacuum block $\sim \langle \mathcal{B}_{ij}^{(1)} \mathcal{B}_{kl}^{(1)} \rangle$ and $\mathcal{K}^{(2)}$ computed from (3.27) using $\langle \epsilon\epsilon\phi\phi \rangle$. The cross ratios we use are

$$z = \frac{z_{12} z_{34}}{z_{13} z_{24}}, \ \xi = \frac{z_{15} z_{34}}{z_{13} z_{54}}, \ \eta = \frac{z_{16} z_{34}}{z_{13} z_{64}}, \ \sigma = \frac{z_{17} z_{34}}{z_{13} z_{74}}, \ \chi = \frac{z_{18} z_{34}}{z_{13} z_{84}}, \ \zeta = \frac{z_{10} z_{34}}{z_{13} z_{04}} \tag{E.1}$$

We first denote the functional value of the 6pt comb channel in terms of the above cross-ratio as

$$V_{h_\phi}^{(6)}(z, \xi, \eta) := \frac{\langle \mathcal{B}_{35}^{(1)} \mathcal{B}_{46}^{(1)} \phi_{1,2} \rangle'}{\langle \phi_{1,2} \rangle}. \tag{E.2}$$

The simpler 8pt comb channel depicted in Figure-2(b) is then given by

$$
\begin{aligned}
V_{T,comb}^{(8),Vir} &= \left. \frac{\langle X_{3,5,7} | \mathbb{I} | \phi_{1,2} | \mathbb{I} | Y_{4,6,8} \rangle}{\langle X_{3,5,7} \rangle \langle \phi_{0,1,2} \rangle \langle Y_{4,6,8} \rangle} \right|_{\mathcal{O}(c^{-2})} \\
&= h_X h_Y \frac{\left\langle \mathcal{B}_{357}^{(1)} \mathcal{B}_{468}^{(1)} \phi_1 \phi_2 \right\rangle'}{\langle \phi_1 \phi_2 \rangle} + h_\phi h_X \frac{\langle \mathcal{B}_{1,2}^{(1)} \mathcal{B}_{3,5,7}^{(1)} Y_{4,6,8} \rangle'}{\langle Y_{4,6,8} \rangle} + h_\phi h_Y \frac{\langle \mathcal{B}_{1,2}^{(1)} \mathcal{B}_{4,6,8}^{(1)} X_{3,5,7} \rangle'}{\langle X_{3,5,7} \rangle}
\end{aligned}
\tag{E.3}
$$

We now separately give the expression for each of the 3 contributions above. The first term can be written as

$$\frac{\left\langle \mathcal{B}_{357}^{(1)} \mathcal{B}_{468}^{(1)} \phi_1 \phi_2 \right\rangle'}{\langle \phi_1 \phi_2 \rangle} = \frac{1}{4} V_{h_\phi}^{(6)}(z, \xi, \eta) + \text{cyc}(3, 5, 7)\text{cyc}(4, 6, 8) = h_\phi V_8(z, \xi, \eta, \sigma, \chi) \tag{E.4}$$

The second term is

$$\frac{\langle \mathcal{B}_{1,2}^{(1)} \mathcal{B}_{4,6,8}^{(1)} X_{3,5,7} \rangle'}{\langle X_{3,5,7} \rangle} = \frac{1}{4} V_{h_X}^{(6)}(1 - \xi, 1 - z, 1 - \eta) + \frac{h_X^2}{8} E^{(7)}(z, \xi, \eta, \sigma) + \text{cyc}(3, 5, 7)\text{cyc}(4, 6, 8) \tag{E.5}$$

$$E^{(7)}(z, \xi, \eta, \sigma) = \langle \mathcal{B}_{1,2}^{(1)} \mathcal{B}_{3,5}^{(1)} \rangle \langle \mathcal{B}_{4,6}^{(1)} (\mathcal{B}_{5,7}^{(1)} + \mathcal{B}_{3,7}^{(1)} - \mathcal{B}_{3,5}^{(1)}) \rangle \tag{E.6}$$

The above expression can be easily computed as $\langle \mathcal{B}_{1,2}^{(1)} \mathcal{B}_{3,5}^{(1)} \rangle$ is just the leading $1/c$ 4pt. vacuum block. Similarly the third term is

$$\frac{\langle \mathcal{B}_{1,2}^{(1)} \mathcal{B}_{3,5,7}^{(1)} Y_{4,6,8} \rangle'}{\langle Y_{4,6,8} \rangle} = \frac{1}{4} V_{h_Y}^{(6)}(\tfrac{1}{\eta}, \tfrac{1}{\xi}, \tfrac{1}{z}) + \frac{h_Y^2}{8} E^{(7)}(\tfrac{z}{z-1}, \tfrac{\eta}{\eta-1}, \tfrac{\xi}{\xi-1}, \tfrac{\chi}{\chi-1}) + \text{cyc}(3, 5, 7)\text{cyc}(4, 6, 8) \tag{E.7}$$

Above we have used the following notation for additional terms obtained by permuting $\text{cyc}(3 \to 5 \to 7)\text{cyc}(4 \to 6 \to 8)$ contributing a total of 9 terms shown below

$$F(z, \xi, \eta, \sigma, \chi) + \text{cyc}(3, 5, 7)\text{cyc}(4, 6, 8) =$$

$$
\begin{aligned}
&= F(z, \xi, \eta, \sigma, \chi) + F\left(\frac{z}{\xi}, \frac{\sigma}{\xi}, \frac{\eta}{\xi}, \frac{1}{\xi}, \frac{\chi}{\xi}\right) + F\left(\frac{z}{\sigma}, \frac{1}{\sigma}, \frac{\eta}{\sigma}, \frac{\xi}{\sigma}, \frac{\chi}{\sigma}\right) \\
&\quad + F\left(\frac{z - \eta z}{z - \eta}, -\frac{(\eta - 1)\xi}{\xi - \eta}, \frac{(\eta - 1)\chi}{\eta - \chi}, \frac{(\eta - 1)\sigma}{\eta - \sigma}, 1 - \eta\right) + F\left(\frac{z(\xi - \eta)}{\xi(z - \eta)}, \frac{\sigma(\xi - \eta)}{\xi(\sigma - \eta)}, \frac{\chi(\xi - \eta)}{\xi(\chi - \eta)}, \frac{\xi - \eta}{\xi - \eta\xi}, 1 - \frac{\eta}{\xi}\right) \\
&\quad + F\left(\frac{z(\sigma - \eta)}{\sigma(z - \eta)}, \frac{\eta - \sigma}{(\eta - 1)\sigma}, \frac{\chi(\sigma - \eta)}{\sigma(\chi - \eta)}, \frac{\xi(\sigma - \eta)}{\sigma(\xi - \eta)}, 1 - \frac{\eta}{\sigma}\right) + F\left(\frac{z - \chi z}{z - \chi}, -\frac{\xi(\chi - 1)}{\xi - \chi}, 1 - \chi, -\frac{\sigma(\chi - 1)}{\sigma - \chi}, -\frac{\eta(\chi - 1)}{\eta - \chi}\right) \\
&\quad + F\left(\frac{z(\xi - \chi)}{\xi(z - \chi)}, \frac{\sigma(\xi - \chi)}{\xi(\sigma - \chi)}, 1 - \frac{\chi}{\xi}, \frac{\xi - \chi}{\xi - \xi\chi}, \frac{\eta(\xi - \chi)}{\xi(\eta - \chi)}\right) + F\left(\frac{z(\sigma - \chi)}{\sigma(z - \chi)}, \frac{\sigma - \chi}{\sigma - \sigma\chi}, 1 - \frac{\chi}{\sigma}, \frac{\xi(\sigma - \chi)}{\sigma(\xi - \chi)}, \frac{\eta(\sigma - \chi)}{\sigma(\eta - \chi)}\right) \tag{E.8}
\end{aligned}
$$

*Note* $\mathcal{B}_{ijk}^{(1)} = \frac{1}{2}(\mathcal{B}_{ij}^{(1)} + \mathcal{B}_{jk}^{(1)} + \mathcal{B}_{ki}^{(1)})$. The terms are generated by considering all possible terms of the form $\langle \mathcal{B}_{ij}^{(1)} \mathcal{B}_{ab}^{(1)} \phi_{1,2} \rangle$ due the definition of $\mathcal{B}_{ijk}^{(1)}$.

For the simpler 9pt function generalization of the Virasoro comb channel in Figure-3(b) we make use of the function

$$B(z,\xi,\eta,\sigma,\chi,\zeta) = V_8(z,\xi,\eta,\sigma,\chi) + \frac{h_\phi}{2}\langle \mathcal{B}_{357}^{(1)} \mathcal{B}_{01}^{(1)} \rangle \left( \langle \mathcal{B}_{468}^{(1)} \mathcal{B}_{02}^{(1)} \rangle + \langle \mathcal{B}_{468}^{(1)} \mathcal{B}_{12}^{(1)} \rangle - \langle \mathcal{B}_{468}^{(1)} \mathcal{B}_{01}^{(1)} \rangle \right). \tag{E.9}$$

The 9pt comb channel is then given by

$$V_{T,comb}^{(9),Vir} = h_X h_Y h_\phi \left[ V_9(z,\xi,\eta,\sigma,\chi,\zeta) + V_9\left( \tfrac{\sigma-\xi}{\zeta-\xi}, \tfrac{\xi}{\xi-\zeta}, \tfrac{\eta-\xi}{\zeta-\xi}, \tfrac{z-\xi}{\zeta-\xi}, \tfrac{\chi-\xi}{\zeta-\xi}, \tfrac{\xi-1}{\xi-\zeta} \right) + \right.$$

$$\left. + V_9\left( \tfrac{(\zeta-1)(\eta-\chi)}{(\eta-1)(\zeta-\chi)}, \tfrac{(\zeta-1)(\eta-\xi)}{(\eta-1)(\zeta-\xi)}, \tfrac{(\zeta-1)\eta}{\zeta(\eta-1)}, \tfrac{(\zeta-1)(\eta-\sigma)}{(\eta-1)(\zeta-\sigma)}, \tfrac{(\zeta-1)(z-\eta)}{(\eta-1)(z-\zeta)}, \tfrac{\zeta-1}{\eta-1} \right) \right] \tag{E.10}$$

where $V_9$ captures the contribution from

$$\frac{\langle X_{3,5,7} | T | \phi_{0,1,2} | T | Y_{4,6,8} \rangle}{\langle \phi_{0,1,2} \rangle \langle X_{3,5,7} \rangle \langle Y_{4,6,8} \rangle} = h_X h_Y \frac{\left\langle \mathcal{B}_{357}^{(1)} \mathcal{B}_{468}^{(1)} \phi_{0,1,2} \right\rangle'}{\langle \phi_{0,1,2} \rangle} = h_X h_Y h_\phi V_9(z,\xi,\eta,\sigma,\chi,\zeta)$$

$$8V_9(z,\xi,\eta,\sigma,\chi,\zeta) = B(z,\xi,\eta,\sigma,\chi,\zeta) + B\left( \frac{z-\zeta}{z-1}, \frac{z-\xi}{z-1}, \frac{z-\eta}{z-1}, \frac{z-\sigma}{z-1}, \frac{z-\chi}{z-1}, \frac{z}{z-1} \right)$$

$$+ B\left( \frac{\zeta}{\zeta-1}, \frac{\zeta-\xi}{\zeta-1}, \frac{\zeta-\eta}{\zeta-1}, \frac{\zeta-\sigma}{\zeta-1}, \frac{\zeta-\chi}{\zeta-1}, \frac{\zeta-z}{\zeta-1} \right) \tag{E.11}$$

The functional form of $B - V_8$ is

$$= -\frac{2}{\zeta^2 z(\eta-\chi)(z-\zeta)}$$

$$\left( \frac{\zeta(\xi+\sigma)\log\left( \frac{\xi(\zeta-\sigma)}{\sigma(\zeta-\xi)} \right)}{\xi-\sigma} + \frac{2\xi\sigma\log\left( \frac{\sigma(\zeta-\xi)}{\xi(\zeta-\sigma)} \right)}{\xi-\sigma} + \frac{((\zeta-2)\xi+\zeta)\log\left( \frac{(\zeta-1)\xi}{\zeta-\xi} \right)}{\xi-1} + \frac{((\zeta-2)\sigma+\zeta)\log\left( \frac{(\zeta-1)\sigma}{\zeta-\sigma} \right)}{\sigma-1} - 6\zeta \right)$$

$$\left( \zeta^2 z^2 \log\left( \frac{(\zeta-\eta)(\chi-z)}{\zeta-\chi} \right) - \eta^2 z^2 \log(\eta-\zeta) - 2\zeta\chi z^2 \log(\eta-\zeta) + 2\zeta\eta z^2 \log\left( \frac{\zeta-\chi}{z-\chi} \right) + 2\eta\chi z^2 \log\left( \frac{\chi(\zeta-\eta)}{\zeta-\chi} \right) \right.$$

$$- \zeta\log(\eta-z)\left( \zeta\eta(\eta-2\chi) + z^2(\zeta-2\chi) - 2\eta z(\eta-2\chi) \right) + 3\zeta\eta z^2 + \chi^2 z^2 \log\left( 1 - \frac{\zeta}{\chi} \right)$$

$$- 3\zeta\chi z^2 + 2\zeta^2\eta\chi \log\left( -\frac{\chi}{z-\chi} \right) - 3\zeta^2\eta z + \zeta^2\chi^2 \log\left( 1 - \frac{z}{\chi} \right) + 3\zeta^2\chi z + 4\zeta\eta\chi z \log\left( 1 - \frac{z}{\chi} \right)$$

$$\left. + \eta(\eta-2\chi)\log(\eta)(z-\zeta)^2 + 2\zeta\chi^2 z \log\left( -\frac{\chi}{z-\chi} \right) \right) \tag{E.12}$$

The functional form $V_{h_\phi}^{(6)}$ can be read from (3.26)

$$V_{h_\phi}^{(6)}(z,\xi,\eta) = h_\phi(h_\phi-1)\langle \mathcal{B}_{35}^{(1)} \mathcal{B}_{12}^{(1)} \rangle \langle \mathcal{B}_{12}^{(1)} \mathcal{B}_{46}^{(1)} \rangle + h_\phi \left( \tfrac{12}{c} \right)^2 \mathcal{K}^{(2)}$$

$$= h_\phi(h_\phi-1)\left( \frac{2(\xi(z-2)+z)}{(\xi-1)z}\log\left( \frac{\xi-z}{\xi-\xi z} \right) + 4 \right)\left( 4 - \left( 2 - \frac{4\eta}{z} \right)\log\left( 1 - \frac{z}{\eta} \right) \right) + h_\phi \left( \tfrac{12}{c} \right)^2 \mathcal{K}^{(2)} \tag{E.13}$$

The functional form of $\mathcal{K}^{(2)}$ is to long and would serve no practical purpose to write it here in its full detail. We note that it consists of 3 terms

$$\mathcal{K}^{(2)} = V_{\text{Li}} + V_{\log^2} + V_{\log} \tag{E.14}$$

where the first is a term containing $\text{Li}_2$, the second and third are quadratic and linear in the Logarithms respectively. The expression for $\mathcal{K}^{(2)}$ is easily obtainable from the expression (3.27).

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
