# Peer review of "Reparametrization mode Ward Identities and chaos in higher-pt. correlators in CFT$_2$"

_SciPost Physics_

## Round 4 · Referee Report · Anonymous (Referee 2) · 2022-10-26

Strengths

1- Presentation has been greatly improved compared to the previous version 2- The authors compute the non-maximal OTO braiding and find no evidence of scrambling whatsoever---an interesting result

Weaknesses

1- Claims made in the paper are not sufficiently justified 2-There remain a few holes in the presentation of the steps 3-Section on 8- and 9-pt functions requires work

Report

In this paper, the authors engage in the theory of reparametrization modes in 2d CFT--denoted $\epsilon(z,\bar{z})$--which encode (multi-)stress-tensor exchanges in the Virasoro identity block of arbitrary numbers of external operators. The paper attempts to understand how Virasoro ward identities govern the form of multiple $\epsilon$ -insertions.

In particular the authors use this construction to compute the leading-order, 5-pt Virasoro comb channel block, which they then use to compute the scrambling behavior of various OTO braidings of operators. The authors find that, for non-maximally braided combinations of operators, there is no scrambling whatsoever---a peculiar result.

There are a few holes in the argument that I think still need to be cleared up. First and foremost, the most important object computed by the authors is denoted $V^{\rm6 (Vir)}_{T, \text{comb}}$ and is given in equation (3.36). It remains unclear to me why this object is the Virasoro contribution to the block at order $c^{-2}$. For example, in 2005.06440, it was shown that the Virasoro contributions to the 6-pt star block come from higher order terms in the expansion in terms of the reparametrization mode $\epsilon$ (contributions to $\mathcal{B}$ at order $\epsilon^2$ or higher). But equation (3.36) restricts only to the leading piece in this $\epsilon$ expansion, and no argument is given for why no additional contribution appear at the same order in $c^{-2}$, e.g. from objects like $\mathcal{B}^{(2)}$. For example, there is an additional contribution to the star channel Virasoro block that appears at order $c^{-2}$ as computed in 2005.06440. The authors should at least explain why this does not happen in the comb channel.

(I suspect that there actually is an additional contribution, which would explain away the lack of scrambling for the non-maximal OTO braiding analyzed in section 4.2.2. But this remains to be seen. )

It would also be useful to know how (3.36) differs from the comb block given in 1810.03244, part of which is given in (4.18). Does the additional Virasoro piece com from $\mathcal{K}^{(2)}$ in (3.27)? Is there any part that looks like the comb block of 1810.03244? Such a consistency check should not be left up to the reader.

My remaining confusion is about section 3.3.1. The graphs depicted in figure 2, to my eyes, do not form a convergent basis of functions for correlators. This is for the simple reason that fusions of triples of operators doesn't allow for arbitrary fusions of two operators in the subgraph. I would thus not call this a comb channel stress tensor block, as pairwise operators of the same dimension can always fuse into $T$. The section may be relevant since the block related to this type of graph is, in fact, computable, but of limited use since the choice of channel is one few would consider.

Requested changes

1- Explain why (3.36) is the honest Virasoro comb block at order $c^{-2}$ or find the additional contribution from $\epsilon^2$ terms.

2- If extra $\epsilon^2$ terms contribute, find how they affect the calculation of the OTOCs

3-Give an account of how (3.36) differs from (4.18)

4-Additional explanations required in 3.3.1 as to the relevance of these types of graphs to physical correlators.

---

## Round 4 · Referee Report · Anonymous (Referee 1) · 2022-10-31

Report

The authors further explore the reparametrization mode representation for stress-tensor exchanges in higher point correlation functions in two dimensional CFT and use these results to study the late time behavior of several out-of-time-order configurations in these higher point functions.

While the work has technical merit and interesting conclusions especially regarding the absence of Lyaponov growth in certain out-of-time-order configurations, I believe that further clarification is needed in order to justify the claims presented.

In particular, one of the primary results is equation (3.36), concerning the ${\cal O}(1/c^2)$ contributions to the Virasoro identity block in the comb channel, but it is not clear to me that this is the only contribution to this block at this order. Naively, I would expect that, representing the Virasoro identity block in the comb channel via projectors as $\langle X_3 X_5 \vert 1 \vert \phi_1 \vert \phi \vert \phi_2 \vert 1 \vert Y_4 Y_6 \rangle$, there would be more contributions at ${\cal O}(1/c^2)$ than the $\langle X_3 X_5 \vert T \vert_g \phi_1 \phi_2 \vert T \vert_g Y_4 Y_6 \rangle$ contribution computed from, e.g., ${\cal O}(1/c^2)$ contributions to either of the projectors $\vert 1 \vert$, $\vert \phi \vert$. As it stands, more justification is required to support (3.36) and consequently the resulting behavior in the various out-of-time-order configurations considered later.

Another minor point concerning the result (3.36) concerns the ambiguities present in the auxiliary function ${\cal A}(z)$ presented in equation (3.21). It is claimed that these ambiguities "do not contribute to anything physical", but I think a little elaboration on this point is warranted. Is the claim that such terms drop out upon constructing the correlator with the bilinears ${\cal B}^{(1)}$?

In addition, I find the discussion of the 8- and 9-point blocks confusing, as the contributions considered don't correspond to any standard conformal block decomposition I'm aware of. Due to the fusion of three operators into the stress tensor, it seems like this is a resummation of a particular subset of a full conformal block decomposition (i.e. conventional conformal block diagrams only have trivalent vertices instead of the 4-point vertices in figures 2, 3), rather than an individual block as referred to in the text. So while the calculation of this contribution may be valid, it is unclear to me how to interpret the result in terms of a systematic expansion of the full correlation function.

---

## Round 4 · Author Response

As mentioned in our previous reply we are able to compute the Virasoro contribution to the 6pt stress-tensor comb channel of 3 pairs of light operators. This we believe has not been computed in literature before. We are able to do this after having worked out the Ward identity associated with $\langle\epsilon\epsilon\phi\phi\rangle$. The analysis of this Ward identity remains unchanged.

We verify that the Virasoro contribution to the 6pt maximally OTOC does exhibit the chaotic behaviour found for the global contribution to the same channel in \ref[36]. We also investigate the 4pt OTO in the 6pt comb channel when the X and Y pairs in Fig-1 are OTOed only w.r.t. each other.

Contrary to our previous reply we find that both the global and Virasoro contributions to the 6pt function do not show an exponential behaviour for such an OTOing. This we find in contrast with the computation for the 6pt star channel in \ref[36] where taking into account the Virasoro contribution gave rise to the exponentially chaotic behaviour.
The understanding arrived at in \ref[36] was that one needs to take into account the gravitational(Virasoro) interaction at the desired order in 1/c which corresponds to taking into account the Virasoro descendants in the internal legs of (star) channel.
In the 6pt comb channel depicted in Fig-1 this amounts to considering the Virasoro descendants of the operator $\phi$ propagating in the internal leg of the channel. However, the global and the Virasoro contributions do not show an exponentially chaotic behaviour. We find this result pertaining to the Virasoro stress-tensor 6pt comb channel intriguing.

Thus, we are able to show that the Ward identity related to $\langle\epsilon\epsilon\phi\phi\rangle$ allows us to compute the Virasoro contribution to the stress-tensor comb channel of 6pt correlator. This Virasoro contribution has a peculiarity that when only the X & Y pairs as depicted in Fig-1 are OTOed w.r.t. each other, the correlator does not show any exponential behaviour.

We thank the referees for insisting on the check of the former claims made in the previous versions of the document with regards to the results obtained in \ref[36]. This -though cumbersome at first, has allowed us to improve the manuscript to a great extent and a better understanding of the physics.

We hope the present version also is more clearer in its explanation and we eagerly await any comments. We apologize for the delay in bringing out the current version of the manuscript.

---

## Round 4 · List of Changes

Below is a short list of changes to sections and subsections. The changes are enumerated according to the (sub,subsub)sections they occur.

Section-2) has two subsections as compared to the older version: subsec
2.1) Deals with the shadow formalism in d dim.
2.2) Deals with the same in the 2d case.

Section-3) has had substantial changes. We claim that the Ward identity with 2 insertions of the reparametrization mode computes the comb channel and not the star channel. The claim of a representation of the projector onto the the Virasoro block using the reparametrization modes has been dropped in its entirety along with the subsequent statements about heavy operators in a correlator.
3.1) Reviews known results for Ward identity associated with single insertion of the reparametrization mode.
3.2) Analyses the Ward identity for 2 insertions of the reparametrization mode into a 2pt function of scalars. The analysis upto eq 3.27 (3.28 in old ver) is unchanged.
3.2.1) The last paragraph from this subsubsection (3.0.1 old ver) has been removed.
3.2)This subsection titled Channel Diagramatics replaces subsec-3.1,,3.2 & 3.3 of the old ver. This section now contains the comb channel computation for 6pt function.
3.2.1) Generalizes the 6pt comb channel to simpler 8pt and 9pt comb channel computations by replacing identical pairs of operators with identical triplets.

Section-4) This section too has substantial changes:
4.1) Contains with few changes and includes the contents up till eq 4.13 of the old version as it still computes the same OTOC but now interpreted for the 6pt comb channel with Virasoro contribution. This OTOC sees the same chaotic behaviour as the global 6pt comb channel unsurprisingly.
4.2) Computes the specified 4pt OTO in the 6pt comb channel
4.2.1) Finds that such a 4pt OTO in the 6pt global comb channel does not show any exponential chaotic behaviour.
4.2.2) Finds that even the Virasoro contribution to the 6pt comb channel for the 4pt OTO does not show any exponential chaotic behaviour. This is in stark contrast with the computation for the 6pt star channel in ref[36] where the inclusion of the Virasoro contribution endows a chaotic behaviour which the global 6pt star channel lacked.

4.3) Finds the chaotic behaviour for a 6pt function by generalizing a 4pt function and finds that such simpler generalizations do not change the chaotic behaviour. Thus suggesting the simpler generalizations to higher pt functions in subsubsec-3.2.1 do not give rise to any different chaotic beahviour than that of the parent 6pt function.

Section-5) Stays unchanged.

Appendices: A, B & C remain unchanged (the contents of the old D & E appendices have been removed)
D) Summarizes the Virasoro comb channel contribution to the 8pt and 9pt functions obtained in subsubsec-3.2.1 in as succinct a form as possible. Any further exposition of the results would lead cumbersome expressions.

---

## Round 5 · Author Response

The channel we end up computing is the stress-tensor comb channel which can be obtained by demanding that 2 of the pairs (of identical operators) in the comb channel fuse to produce the stress-tensor. This uniquely fixes the operator propagating in the remaining internal leg (c.f. Fig-1). We term this the stress-tensor contribution to the 6pt comb channel (defined in eq(3.37)) which we compute to $\mathcal{O}(1/c^2)$.
The changes to the manuscript with relevant comments are as follows:
1) The value of $\mathcal{A}$ can be shifted by a function of the form in eq(3.22) and this would not effect the final form of $\mathcal{K}^{(2)}$ computed in eq(3.27). $c.f.$ footnote-16 on pg10. This has been checked explicitly.
2) We check for additional contributions to the above stress-tensor Virasoro comb channel at $\mathcal{O}(1/c^2)$ by inserting the projector onto the Virasoro descendants of the vacuum (eq(3.38) and Appendix-D) between the pair of identical operators. We find that the additional contribution simply symmetrizes the answer w.r.t. the 3 pairs of operators. The final answer is eq(3.41). This contribution can be interpreted as that coming from $\epsilon^2$ as the additional contributions are similar in form to eq(3.36).
3) We take into the additional contributions to the Virasoro stress-tensor comb channel in computing the OTOC of the kind where only the X and Y pairs are out of time ordered (section 4.2) and still find no exponential growth.
4) We cannot see the global stress-tensor comb channel eq(4.18) in any obvious form within the final answer eq(3.41) or in a part of it like eq(3.36). We state this below eq(4.19). It could be due to the fact that the method used to compute the full Virasoro contribution (Fig-1b) makes use of the 2pt function of $\phi$ while the global contribution (Fig-1a) computed for example in 2005.06440 (c.f. eq (5.1)of 2205.06440) explicitly makes use of the projector onto the global descendants of $\phi$.
5) We mention that the computation of simpler higher pt functions are done since the method allows them (beginning of section 3.3.1). These higher pt. functions have 4pt vertices which we show (pg-17, Fig-4) can be decomposed into a diagram consisting of 2 3pt vertices and an internal exchange of an additional stress-tensor. We also argue that if this susbtitution is made into the 4pt vertices of the higher pt comb channel we get stress-tensor comb channels where the operators propagating in the internal legs are fixed by demanding that the like operator (pairs) on either ends fuse to give the stress-tensor.
6) We also appropriately correct the answers for the Virasoro stress-tensor comb channel for these higher pt. functions considered in subsection-3.3.1 by accounting for all $\mathcal{O}(1/c^2)$ corrections in a manner similar to the 6pt case in eq-3.57 & eq-3.58 and Appendix-E.

---

## Round 5 · List of Changes

1) Footnote-16 on pg-10. 2) Footnote-19 on pg-11 explaining the relation between $\mathcal{K}^{(2)}$ and $\mathcal{C}^{(2)}$ better in eq-3.27 3) New equations for checking all relevant $1/c^{2}$ contributions from eq-3.27 to eq-3.41 along with Appendix-D. 4) New expressions accounting corrections to simpler higher-pt stress-tensor comb channels eq-3.57 & eq-3.58 and relevant correction in Appendix-E. 5) Comments and a new Fig-4 below eq-3.58 explaining a possible way to split 4pt vertex into one consisting 2 3pt vertex. 6) Comments below eq-4.13. 7) Comments below eq-4.19 explaining how expression in eq-4.18 (Global T comb-channel) is not obvious in eq-3.41 (Virasoro T comb channel). 8) Eq-4.20, eq-4.23 & eq-4.24 and relevant comments accounting for the corrections due to eq-3.41.

---

## Editorial Decision

unknown